# NEUQI: NEAR-OPTIMAL UNIFORM QUANTIZATION PARAMETER INITIALIZATION

## ABSTRACT

Large language models (LLMs) achieve impressive performance across domains but face significant challenges when deployed on consumer-grade GPUs or personal devices such as laptops, due to high memory consumption and inference costs. Post-training quantization (PTQ) of LLMs offers a promising solution that reduces their memory footprint and decoding latency. In practice, PTQ with uniform quantization representation is favored due to its efficiency and ease of deployment, as uniform quantization is widely supported by mainstream hardware and software libraries. Recent studies on $\geq$ 2-bit uniform quantization have led to noticeable improvements in post-quantization model performance; however, they mainly focus on quantization methodologies, while the initialization of quantization parameters remains underexplored and still relies on the conventional Min-Max strategic rationale. In this work, we identify the limitations of the Min-Max strategic rationale, move beyond its constraints, and propose **NeUQI**, a method that efficiently determines near-optimal initialization for uniform quantization. Our NeUQI proposes a method that determines the near-optimal zero-point for a given scale, thereby reformulating the initialization optimization into a scale-only problem that can be solved efficiently. Benefiting from the improved quantization parameters, our NeUQI consistently outperforms existing methods in the experiments with the LLaMA and Qwen families on various settings and tasks. Furthermore, when combined with a lightweight distillation strategy, NeUQI even achieves superior performance to PV-tuning, a considerably more resource-intensive method.

## 1 INTRODUCTION

In recent years, large language models (LLMs) like ChatGPT (OpenAI et al., 2024) have rapidly emerged, demonstrating strong capabilities across various tasks, including open-ended writing, knowledge-based question answering, and code generation. Given the high API costs of proprietary LLMs and the growing performance of open-source alternatives like LLaMA (Touvron et al., 2023; Grattafiori et al., 2024) and Qwen (Yang et al., 2025) families, there is a rising preference for deploying open-source LLMs locally. However, the deployment of large-scale models (e.g., LLaMA 3 70B) is often limited by compute resources and inference efficiency, especially on personal devices or consumer-grade GPUs like a single RTX 4090 with 24GB of memory. In this context, post-training quantization (PTQ) (Krishnamoorthi, 2018), particularly using a uniform quantization scheme, offers a practical solution by converting model weights from `bfloat16` to low-bit-width integer formats such as `int4/3/2`, significantly reducing memory usage and inference latency.

In the context of uniform quantization, the Min-Max initialization method (Jacob et al., 2017) is simple and effective at higher bit-widths (e.g., 8-bit, 4-bit) (Dettmers et al., 2022; Frantar et al., 2023), but its simplistic design becomes ineffective in lower bit-width settings such as 2-bit and 3-bit. Although prior works on $\geq 2$ bits uniform PTQ have achieved notable progress, they still fundamentally rely on the Min-Max strategic rationale for initialization. This strategic rationale has two key specifications that lead to two limitations, as discussed in Section 3.3: (1) the determination of both the scale and zero-point by the extreme values of the vector constrains the design of the optimization algorithm, and (2) the integer zero-point constraint restricts the parameter space. The drawback of the first specification is most pronounced in its unnecessary expansion of the search space for search-based methods such as LeanQuant (Zhang & Shrivastava, 2025).

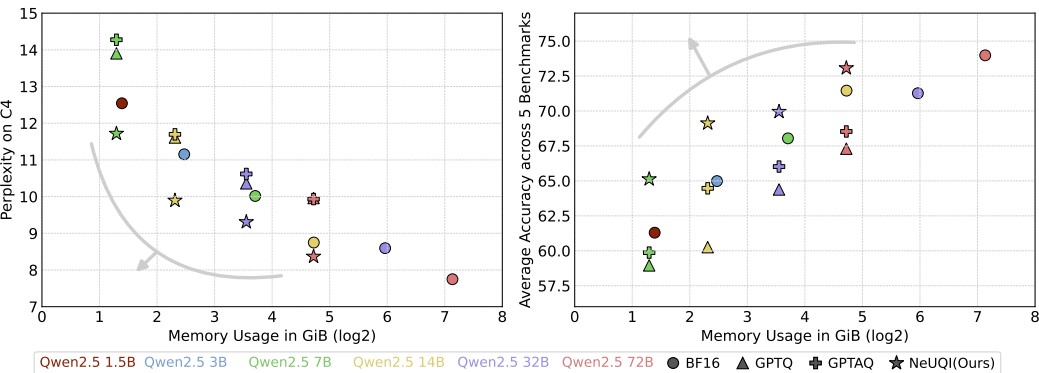

Figure 1: Perplexity on C4 (left) and average accuracy across five common benchmarks (right) with Qwen 2.5 family, plotted against the base-2 logarithm ($\log_2$) of model memory usage on the x-axis. Gray arrows indicate the direction of better performance, corresponding to lower perplexity and higher accuracy with smaller model memory usage. The results include non-quantized models (BF16) and quantized models at 3-bit using GPTQ, GPTAQ, as well as our proposed NeUQI. More details are provided in Section 5.

Recognizing the limitations of the two preceding specifications, we depart from them and exploit the modified structure of the loss function to develop an efficient method, NeUQI. To address the joint optimization of the scale and zero-point, we propose an efficient method that computes a near-optimal zero-point for a given scale, reducing the problem to a single-variable optimization over the scale, which is efficiently solved via a coarse-to-fine grid search. Empirical results show that our NeUQI substantially improves the performance of quantized models and, under the 3-bit channel-wise setting, can even surpass that of non-quantized models with comparable memory usage, as illustrated in Figure 1. By eschewing the Min–Max strategic rationale in determining quantization parameters, NeUQI can provide effective initialization that not only enables lightweight distillation to surpass PV-tuning (Malinovskii et al., 2024), a more sophisticated and resource-intensive fine-tuning method, but also yields further improvement when adopted in the strong fine-tuning method EfficientQAT (Chen et al., 2025). These findings underscore the critical role of better initialization: poor initial conditions are difficult to fix even with extensive fine-tuning, whereas a well-initialized model can achieve better performance with significantly less effort.

The main contributions of our work are summarized as follows:

1. We identify two specifications within the conventionally adopted Min-Max strategic rationale for uniform PTQ initialization, which also represent its limitations, thereby contributing to a deeper understanding of initialization strategies and revealing a feasible direction for future research.

2. To move beyond the Min-Max strategic rationale and confront the corresponding altered structure of the loss function, we propose NeUQI, a novel and efficient initialization method for uniform quantization.

3. Extensive experiments on the LLaMA and Qwen model families across various sizes and tasks demonstrate that our NeUQI consistently outperforms existing methods. Furthermore, experiments on fine-tuning methods such as PV-tuning and EfficientQAT highlight the importance of better initialization and the effectiveness of NeUQI.

## 2 RELATED WORK

**Uniform Quantization** In the 2-bit to 4-bit quantization regime, prior studies have explored a range of post-training quantization techniques to mitigate accuracy degradation. Early methods such as SmoothQuant (Xiao et al., 2023) and AWQ (Lin et al., 2024b) applied channel-wise transformations, shrinking activations while compensating by enlarging the corresponding weights. Subsequent works extended channel-wise scaling into efficient invertible transformations for improved smoothing distribution, as exemplified by QuIP (Chee et al., 2023), DuQuant (Lin et al.,

2024a), FrameQuant (Adepu et al., 2024), and QuaRot (Ashkboos et al., 2024). Among these, the Hadamard transform, widely adopted as a representative transformation-based technique, offers simplicity while maintaining comparable effectiveness, as demonstrated in QuaRot. Alternatively, MagR (Zhang et al., 2024) suppresses the magnitude of weights while preserving model functionality. In addition, fine-tuning-based approaches have been explored, such as CBQ (Ding et al., 2025), which performs block-wise tuning, and PV-Tuning (Malinovskii et al., 2024), which jointly optimizes continuous and discrete parameters. Meanwhile, SpinQuant (Liu et al., 2025) and OSTQuant (Hu et al., 2025) incorporate learnable transformations into the model, substantially reducing runtime overhead.

**Quantization Parameter Initialization**  Most uniform quantization methods use the conventional Min-Max initialization, while the rest employ variants such as clipping-based, quantile-based, MSE-based, and MeanStd-based approaches (Li et al., 2021). Among them, the MSE-based variant is search-based, whereas the others rely on empirical formulas to estimate the scale and zero-point. The following methods inherit the first specification and its associated limitation of the Min-Max strategic rationale. The fine-tuning-based method OmniQuant (Shao et al., 2024) extends clipping-based Min-Max into learnable weight clipping during training; however, it cannot leverage results obtained by non-fine-tuning methods. Although LeanQuant (Zhang & Shrivastava, 2025) determines parameters through a loss-aware search rather than directly applying Min-Max or its variants, it still suffers from the same limitation, thereby constraining the design of the optimization algorithm.

## 3 BACKGROUND

### 3.1 UNIFORM QUANTIZATION

Quantization is essentially the process of mapping floating-point values to a set $\mathcal{Q}$ of quantized values, where each floating-point number is represented by a low-bit-width index, thereby reducing both memory footprint and memory access latency. In the case of **uniform quantization**, also known as **asymmetric affine quantization**, $\mathcal{Q}$ is defined as a set of equally spaced points. Specifically, $\mathcal{Q}$ is characterized by three parameters: the scale $s$, the zero-point $z$, and the bit-width $k$. Formally, $\mathcal{Q}$ is given by:

$$\mathcal{Q} = \left\{ q_i = s \cdot (i - z) \,\middle|\, i \in \{0, 1, \ldots, 2^k - 1\} \right\}, \tag{1}$$

where $i$ denotes the index of the quantized value, and $q_i$ is the quantized value in $\mathcal{Q}$ corresponding to index $i$. For **symmetric affine quantization**, the zero-point $z$ is fixed to $2^{k-1}$.

To reduce quantization error, each floating-point value is typically mapped to the nearest value in $\mathcal{Q}$. Given a floating-point input $x$, and assuming the parameters $s$, $z$, and $k$ of $\mathcal{Q}$ are known, the uniform quantization function $Q$ is defined as:

$$Q_{s,z}(x) = s \cdot \left( \text{clip}(\lfloor x/s + z \rceil, \, 0, \, 2^k - 1) - z \right), \tag{2}$$

where $\lfloor \cdot \rceil$ is the rounding operator, $\text{clip}(\cdot, a, b) = \max(a, \min(\cdot, b))$ restricts the value to $[a, b]$, and the subscript $k$ is usually omitted. When the input is a vector or matrix, $Q$ is applied element-wise. Notably, in conventional settings (Krishnamoorthi, 2018; Jacob et al., 2017) and prior works, $z$ is placed outside the rounding operation, which restricts $z$ to integer values. In contrast, placing $z$ inside the rounding, as shown above, makes it clear that **z can take floating-point values**.

### 3.2 MIN-MAX INITIALIZATION

Min-Max determines the quantization parameters by mapping the minimum and maximum of the parameter vector $x$ to the minimum and maximum of quantized values. Based on the extremal-value mapping under the conventional setting of the zero-point as a $k$-bit unsigned integer, the **Min-Max formula** is given as follows, from which the scale $s$ and the zero-point $z$ are directly determined:

$$s = \frac{x_{max} - x_{min}}{2^k - 1}, \quad z = -\left\lfloor \frac{x_{min}}{s} \right\rceil. \tag{3}$$

## 3.3 LIMITATIONS WITHIN MIN-MAX

The first specification of the Min-Max strategic rationale requires the scale and zero-point to be determined by the extreme values, which leads to the following limitations. In search-based methods such as LeanQuant, initialization of uniform quantization determines the optimal scale and zero-point by performing grid search over candidate Min-Max value pairs. In LeanQuant, the grid search size is $T$ (typically 2048) for scale and zero-point, yielding a total search space of $T^2$. In contrast, enumerating directly in the scale and zero-point spaces yields $T$ candidates for the scale and $2^k$ candidates for the zero-point (Jacob et al., 2017), where $k \leq 4$ in common practice, resulting in a search space of $2^k T$, which is significantly smaller. Moreover, OmniQuant and SpinQuant also conform to this specification and thus cannot exploit the favorable structures established by non-finetuning PTQ methods. Our lightweight distillation, by contrast, effectively leverages these results. As shown in Table 5, GPTQ with our lightweight distillation matches OmniQuant, while NeUQI with our lightweight distillation even surpasses PV tuning.

The second specification requires the zero-point to be a $k$-bit unsigned integer in $k$-bit quantization, which imposes a significant limitation by reducing the quantization parameter space and constraining achievable performance, particularly in low-bit-width settings. In contrast, NeUQI relaxes this specification and consequently achieves superior performance, as shown in Section 5.3 and Section 5.4, with only a marginal increase in average bit-width. Table 4 further highlights that NeUQI attains better results than methods constrained by this specification, even when operating at a lower average bit-width. For details on hardware support for floating-point zero-points, see Appendix E.

## 4 METHODOLOGY

In this section, we present **NeUQI** (**Ne**ar-Optimal **U**niform **Q**uantization **I**nitialization), a novel method designed to improve uniform quantization initialization. Our NeUQI moves beyond the two specifications of the Min-Max strategic rationale, enabling more effective direct optimization of the scale and zero-point. In the following, we first revisit the formulation of quantization loss, then explain how NeUQI performs the optimization, and finally introduce a lightweight distillation strategy that further enhances the performance of the quantized model with minimal overhead.

### 4.1 QUANTIZATION LOSS

Following the commonly used formulation in GPTQ (Frantar et al., 2023), the quantization loss for a single linear layer is defined with optimization variables $s$ and $z$, as our method dedicates to initializing these quantization parameters, and is given by

$$
\begin{aligned}
\mathcal{L}(s, z) &= \|\boldsymbol{X}(Q_{s,z}(\boldsymbol{W}) - \boldsymbol{W})^\top\|_F^2 \\
&= \operatorname{tr}\left((Q_{s,z}(\boldsymbol{W}) - \boldsymbol{W})\boldsymbol{H}(Q_{s,z}(\boldsymbol{W}) - \boldsymbol{W})^\top\right) \\
&= \sum_i (Q_{s,z}(\boldsymbol{W}_{i,:}) - \boldsymbol{W}_{i,:})\boldsymbol{H}(Q_{s,z}(\boldsymbol{W}_{i,:}) - \boldsymbol{W}_{i,:})^\top,
\end{aligned}
\tag{4}
$$

where $\boldsymbol{X} \in \mathbb{R}^{T \times N}$ denotes the input activations, $\boldsymbol{W} \in \mathbb{R}^{M \times N}$ is the weight matrix of the linear layer, with $N$ and $M$ representing the input and output dimensions respectively. $\boldsymbol{H} = \boldsymbol{X}^\top \boldsymbol{X} \in \mathbb{R}^{N \times N}$ is commonly referred to as the Hessian matrix. This formulation naturally suggests adopting the row-wise quantization strategy, which aligns with common practices that treat one row as one or multiple parameter vectors for quantization.

### 4.2 NEUQI

Although the full Hessian matrix can be efficiently computed for a single linear layer, incorporating it into optimization is nontrivial. To address this, we adopt the diagonal approximation (LeCun et al., 1989; Kingma & Ba, 2017), which removes cross-weight interactions and yields a simpler loss function while remaining globally coupled through shared quantization parameters. Under this approximation, the loss reduces to:

$$
\mathcal{L}(s, z) = \sum_i H_{i,i}(Q_{s,z}(w_i) - w_i)^2.
\tag{5}
$$

To address the coupled optimization of the scale and zero-point, we first fix the scale and propose a method that efficiently computes a near-optimal zero-point. This reduces the original two-variable

problem into a single-variable optimization over the scale, which is efficiently solved using a grid search strategy. To further reduce computation time while maintaining accuracy, we employ a coarse-to-fine grid search strategy.

### 4.2.1 OPTIMIZATION OF ZERO-POINT

Fixing the scale, with $x_i = w_i/s$ and $h_i = H_{i,i}s^2$, the loss function becomes

$$\mathcal{L}(z) = \sum_{i=1}^{n} h_i \big(x_i + z - \text{clip}(\lfloor x_i + z \rceil, 0, 2^k - 1)\big)^2. \tag{6}$$

For sample $i$, the per-sample loss function is

$$\mathcal{L}_i(z) = h_i \big(\text{clip}(\lfloor x_i + z \rceil, 0, 2^k - 1) - (x_i + z)\big)^2 \tag{7}$$

$$= \begin{cases} h_i(x_i + z - 0)^2, & z < \frac{1}{2} - x_i, \\ h_i(x_i + z - j)^2, & j - \frac{1}{2} - x_i \le z < j + \frac{1}{2} - x_i, \quad j = 1, \ldots, 2^k - 2, \\ h_i(x_i + z - (2^k - 1))^2, & z \ge 2^k - \frac{3}{2} - x_i. \end{cases} \tag{8}$$

Hence, $\mathcal{L}_i(z)$ and $\mathcal{L}(z)$ are piecewise quadratic functions of $z$. To determine the global minimum of $\mathcal{L}(z)$, it is necessary to obtain the explicit quadratic function on each interval. As shown in Equation 7, the transition points of $\mathcal{L}_i(z)$ consist of the $2^k - 1$ points $j + \frac{1}{2} - x_i$, for $j = 0, \ldots, 2^k - 2$, which divide the real line into $2^k$ intervals. The transition points of $\mathcal{L}(z)$ are the union of those from all per-sample losses. For analytical and computational purposes, coincident transition points are treated as distinct by inserting zero-length intervals, thereby preserving the function values and leaving the outcome unchanged. As a result, $\mathcal{L}(z)$ has $n(2^k - 1)$ transition points and $n(2^k - 1) + 1$ intervals. A naive method that iterates over each interval and computes the sum of the corresponding per-sample quadratic functions on that interval yields a time complexity of $O(n^2 2^k)$, which is prohibitively large.

However, we observe that between two adjacent intervals of $L(z)$ only one sample function undergoes a change. Suppose the function that changes at the transition point $j + \frac{1}{2} - x_i$ corresponds to sample $i$, thus the function change $\delta_{i,j}(z)$ between

---

**Algorithm 1** Optimal Zero-point over Piecewise Quadratic Function

---

**Require:** Samples $\{x_i\}_{i=1}^{n}$, bit-width $k$
**Ensure:** Optimal zero-point $z^*$ minimizing simplified quantization loss
1: Initialize list of transition points: $\mathcal{T} \leftarrow []$
2: **for** each sample $x_i$ **do**
3:     **for** $j = 0$ to $2^k - 2$ **do**
4:         Transition point: $t \leftarrow j + \frac{1}{2} - x_i$
5:         Loss Function increment: $\delta_{i,j}(z) \leftarrow h_i(x_i + z - (j+1))^2 - h_i(x_i + z - j)^2$
6:         Add $(t, \delta_{i,j}(z))$ to $\mathcal{T}$
7:     **end for**
8: **end for**
9: Sort $\mathcal{T}$ by transition point $t$
10: Get interval-wise loss function: $\mathcal{L}^{\text{I}}(z) \leftarrow \sum_i h_i(x_i + z - 0)^2$
11: **for** each $(t, \delta(z))$ in $\mathcal{T}$ **do**
12:     $\mathcal{L}^{\text{I}}(z) \leftarrow \mathcal{L}^{\text{I}}(z) + \delta(z)$
13:     Let next transition point be $t'$ (or $+\infty$ if none)
14:     Get minimum: $z' \leftarrow \arg\min_{z \in [t,t']} \mathcal{L}^{\text{I}}(z)$
15:     Evaluate $\mathcal{L}^{\text{I}}(z)$ at $z'$, update $z^*$ if smaller
16: **end for**
17: **return** $z^*$ with minimal loss

---

the two adjacent quadratic pieces is $\delta_{i,j}(z) = h_i(x_i + z - (j+1))^2 - h_i(x_i + z - j)^2$. When the intervals are enumerated in increasing order, the quadratic function on the current interval is therefore obtained by updating the result from the previous interval with the single changing term, rather than recomputing the full sum. In this way, the overall complexity of obtaining the quadratic function on every interval of $\mathcal{L}(z)$ reduces to $O(n2^k \log(n2^k))$, since sorting all transition points dominates the total running time. The detailed procedure is provided in Algorithm 1.

However, the resulting computational burden remains substantial. To address this, we propose a two-stage efficient approximation strategy, which reduces the overall time complexity to $O(n \log n)$. More details are provided in Appendix C.

- **Interval Estimation**: At this stage, we approximate $x_i + z$ within the low-loss interval $[-\frac{1}{2}, 2^k - \frac{1}{2}]$ by its maximum value $\frac{1}{4}$, while leaving the quantization loss unchanged outside the low-loss interval, resulting in a smoothed objective:

$$\max\left(\frac{1}{4}, \big(x_i + z - \text{clip}\big(\lfloor x_i + z \rceil, 0, 2^k - 1\big)\big)^2\right). \tag{9}$$

Under this approximation, each sample reduces to exactly two transition points, located at $\frac{1}{2} - x_i$ and $2^k - \frac{1}{2} - x_i$. Leveraging the same technique, we can determine the optimal zero-point $z_{\text{opt}}^{\text{smooth}}$ of the smoothed objective in $O(n \log n)$ time.

- **Interval Traversal**: The original, non-approximated loss function is then evaluated within a narrow interval $[z_{\text{opt}}^{\text{smooth}} - 1, z_{\text{opt}}^{\text{smooth}} + 1]$. As the number of transition points in this interval remains limited to at most two per sample, this step also maintains a time complexity of $O(n \log n)$.

### 4.2.2 Optimization of Scale

After determining the near-optimal zero-point $z^*(s)$ for a given scale $s$ using the above algorithm, the loss function can be simplified into a single-variable function over $s$. Although the loss for some rows in the weight matrix appears unimodal, many other rows exhibit non-unimodal behavior, as illustrated in Figure 2. Specifically, as shown in the figure, the loss landscape for the 1st row has a single minimum, indicating unimodal behavior, whereas the 3709th rows each exhibit two local minima, clearly showing non-unimodal patterns. As a result, it is challenging to apply continuous optimization techniques like gradient descent or even one-dimensional search methods like ternary search for selecting the optimal scale $s$.

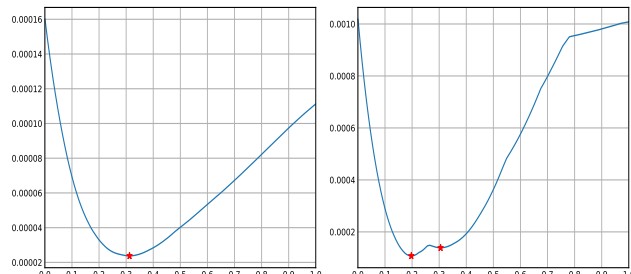

Figure 2: Line plots of $\mathcal{L}(s, z^*(s))$ for the query projection matrix in the 15th transformer layer. The horizontal axis shows the ratio of $s$ to the baseline value $(x_{max} - x_{min})/(2^k - 1)$. The left and right plots correspond to the 1st and 3709th rows of the query projection matrix. Local minima are indicated with red star (★).

For such a continuous function, the most effective approach is to perform an efficient search over evenly spaced points in the parameter space to ensure sufficient scale exploration. Specifically, the candidate scales are defined as follows:

$$\mathcal{S}_T = \left\{ \frac{x_{max} - x_{min}}{2^k - 1} \cdot \frac{i}{T} \,\middle|\, i \in \{1, 2, \ldots, T\} \right\}, \tag{10}$$

where $T$ denotes the total number of candidate scales. We evaluate the quantization error at each candidate scale and select the one with the minimum error as the optimal solution.

To further improve search efficiency, we adopt a coarse-to-fine strategy. We first search on $\mathcal{S}_{T_c}$ to identify a promising region, then refine the search by searching over a local candidate set around the optimal solution $s_c^*$ from $\mathcal{S}_{T_c}$, which is constructed by selecting the $\lfloor T/(2T_c) \rfloor$ nearest neighbors on each side of $s_c^*$ from the finer space $\mathcal{S}_T$, together with $s_c^*$ itself. This yields a total of $2\lfloor T/(2T_c) \rfloor + 1$ scales, enabling more accurate estimation with relatively low overhead.

This strategy effectively reduces the original $T$ expensive error evaluations, each requiring determination of the minimum loss and its corresponding zero-point, to approximately $T_c + T/T_c$. In particular, when $T_c$ is set approximately to $\sqrt{T}$, the total number of evaluations further decreases to around $2\sqrt{T}$. As a result, this approach significantly reduces computational overhead while providing a good approximation to the optimal solution and achieving a favorable trade-off between quantization accuracy and exploration efficiency.

### 4.3 Lightweight Distillation

To better evaluate the effectiveness of NeUQI, we conduct additional lightweight distillation experiments within a fine-tuning framework. Unlike OmniQuant, we treat scale and zero-point as continuous parameters and directly optimize them. During distillation, only these parameters and the RMSNorm layers are updated, while the language model head and embedding layer are kept in original precision to avoid bias and reduce memory usage. The distillation loss is defined as the mean squared error between the final-layer hidden states of the original and quantized models.

Table 1: Perplexity on Wiki2 and C4, and zero-shot accuracy on five benchmarks (averaged as **Acc**) for 2-bit channel-wise quantized models. ↑/↓ indicate whether higher or lower is better, and **Size** denotes the pre-quantization model size. † marks results directly from Zhang & Shrivastava (2025), and ‡ marks results without extra coordinate descent iterations used in their original paper for the fair comparison.

| Model | Size | Bits | Method | Wiki2↓ | C4↓ | ArcC↑ | ArcE↑ | HellaS↑ | PiQA↑ | WinoG↑ | Acc↑ |
|---|---|---|---|---|---|---|---|---|---|---|---|
| LLaMA 2 | 7B | 2 | GPTQ | 6953 | 2592 | 21.93 | 25.63 | 25.89 | 52.23 | 49.72 | 35.08 |
| | | | GPTAQ | 1269 | 246 | 21.76 | 26.89 | 25.74 | 53.32 | 47.12 | 34.97 |
| | | | OmniQuant† | 37.37 | 90.64 | 21.76 | 37.42 | 29.59 | 57.18 | 51.93 | 39.58 |
| | | | MagR‡ | 129 | 47.55 | 21.42 | 34.97 | 32.29 | 58.27 | 50.75 | 39.54 |
| | | | NeUQI | **17.14** | **17.50** | **23.98** | **51.73** | **36.04** | **65.89** | **58.56** | **47.24** |
| | 13B | 2 | GPTQ | 1735 | 433 | 21.76 | 26.52 | 25.85 | 52.56 | 51.30 | 35.60 |
| | | | GPTAQ | 145 | 62.05 | 20.56 | 26.89 | 26.99 | 53.43 | 51.85 | 35.95 |
| | | | OmniQuant† | 17.21 | 26.76 | 24.66 | 48.19 | 40.16 | 63.00 | 52.33 | 45.67 |
| | | | MagR‡ | 47.98 | 27.94 | 23.21 | 26.68 | 36.70 | 52.99 | 54.14 | 38.74 |
| | | | NeUQI | **13.72** | **14.39** | **26.19** | **55.18** | **37.37** | **65.94** | **59.12** | **48.76** |
| | 70B | 2 | GPTQ | 60.29 | 46.11 | 19.97 | 28.07 | 27.05 | 53.75 | 50.04 | 35.78 |
| | | | GPTAQ | 40.09 | 28.37 | 21.59 | 28.91 | 28.98 | 54.08 | 51.46 | 37.00 |
| | | | OmniQuant† | 7.81 | 12.28 | – | – | – | – | – | – |
| | | | MagR‡ | 68.62 | 13.95 | 39.59 | 71.21 | 52.81 | 76.33 | 69.06 | 61.80 |
| | | | NeUQI | **7.03** | **8.88** | **44.71** | **75.97** | **53.62** | **77.64** | **73.72** | **65.13** |
| LLaMA 3 | 8B | 2 | GPTQ | >1e4 | >1e4 | 20.90 | 23.91 | 26.04 | 53.48 | 48.86 | 34.64 |
| | | | GPTAQ | >1e4 | 4409 | 20.99 | 25.29 | 25.88 | 52.99 | 48.46 | 34.72 |
| | | | MagR | 387 | 140 | 18.43 | 30.09 | 27.60 | 55.98 | 50.12 | 36.45 |
| | | | NeUQI | **64.47** | **39.41** | **24.83** | **52.10** | **37.50** | **63.00** | **59.04** | **47.30** |
| | 70B | 2 | GPTQ | >1e4 | >1e4 | 20.73 | 25.76 | 25.76 | 52.45 | 48.46 | 34.63 |
| | | | GPTAQ | >1e4 | >1e4 | 22.70 | 24.87 | 25.78 | 53.21 | 51.22 | 35.56 |
| | | | MagR | >1e4 | >1e4 | 21.59 | 25.59 | 25.51 | 52.99 | 48.46 | 34.83 |
| | | | NeUQI | **56.21** | **42.06** | **25.68** | **53.24** | **36.98** | **61.04** | **55.49** | **46.49** |
| Qwen 2.5 | 7B | 2 | GPTQ | 2332 | 719 | 21.42 | 26.52 | 25.70 | 50.92 | 48.93 | 34.70 |
| | | | GPTAQ | 637 | 244 | 23.21 | 25.34 | 26.49 | 52.23 | 46.96 | 34.85 |
| | | | MagR | **25.38** | **24.83** | **23.12** | 42.09 | 37.76 | **60.99** | 51.30 | 43.05 |
| | | | NeUQI | 37.46 | 28.77 | 22.10 | **47.85** | **39.56** | **60.99** | **58.48** | **45.80** |
| | 14B | 2 | GPTQ | 3852 | 1056 | 23.12 | 25.42 | 25.90 | 50.92 | 51.93 | 35.46 |
| | | | GPTAQ | 1363 | 211 | 22.35 | 25.63 | 25.6 | 51.31 | 49.8 | 34.94 |
| | | | MagR | **18.36** | 19.36 | 23.21 | 44.74 | 38.06 | 63.44 | 52.88 | 44.47 |
| | | | NeUQI | 23.58 | **18.38** | **37.88** | **69.32** | **46.86** | **72.31** | **67.40** | **58.76** |
| | 32B | 2 | GPTQ | 308 | 141 | 21.42 | 26.09 | 25.11 | 51.20 | 49.41 | 34.65 |
| | | | GPTAQ | 277 | 86.61 | 20.82 | 23.86 | 27.08 | 52.23 | 51.22 | 35.04 |
| | | | MagR | **13.16** | **14.72** | 25.94 | 48.91 | 45.25 | 68.06 | 56.51 | 48.93 |
| | | | NeUQI | 18.17 | 15.77 | **41.55** | **71.63** | **53.38** | **76.01** | **72.77** | **63.07** |
| | 72B | 2 | GPTQ | 1127 | 281 | 22.95 | 25.00 | 25.46 | 51.31 | 48.70 | 34.68 |
| | | | GPTAQ | 576 | 105 | 21.59 | 23.40 | 27.87 | 51.74 | 49.09 | 34.74 |
| | | | MagR | 19.25 | 13.83 | 31.91 | 59.64 | 51.13 | 72.74 | 59.59 | 55.00 |
| | | | NeUQI | **10.79** | **11.36** | **48.04** | **78.11** | **56.18** | **78.56** | **75.14** | **67.21** |

# 5 EXPERIMENT

## 5.1 BASELINES AND EVALUATION

We experiment with NeUQI on three commonly-used LLM families, covering different sizes: LLaMA 2 (7B, 13B, 70B) (Touvron et al., 2023), LLaMA 3 (8B, 70B) (Grattafiori et al., 2024) and Qwen 2.5 (7B, 14B, 32B, 72B) (Yang et al., 2025). In the context of **uniform quantization**, we compare with several representative weight-only post-training methods that follow this scheme, including GPTQ (Frantar et al., 2023), MagR (Zhang et al., 2024), GPTAQ (Li et al.,

Table 2: Results of LLaMA 2 7B with W2A16, W2A4, and W4A4 under the Hadamard transform.

| Setting | W2A16 | | | W2A4 | | | W4A4 | | |
|---|---|---|---|---|---|---|---|---|---|
| Method | Wiki2↓ | C4↓ | Acc↑ | Wiki2↓ | C4↓ | Acc↑ | Wiki2↓ | C4↓ | Acc↑ |
| GPTQ | 759 | 277 | 34.95 | 1098 | 463 | 34.88 | 6.14 | 7.75 | 61.67 |
| GPTAQ | 64.88 | 39.36 | 37.14 | 85.65 | 50.17 | 36.78 | 6.20 | 7.73 | 61.93 |
| MagR | 13.79 | 15.05 | 48.01 | 20.42 | 20.54 | 42.95 | 6.02 | 7.59 | 62.69 |
| NeUQI | **12.41** | **13.22** | **52.91** | **13.63** | **14.91** | **50.33** | **5.99** | **7.57** | **62.77** |

2025), OmniQuant (Shao et al., 2024), and LeanQuant (Zhang & Shrivastava, 2025), together with three initialization strategy baselines for analytical purposes. We compare with PV-tuning (Malinovskii et al., 2024) and EfficientQAT (Chen et al., 2025) to analyze the effect of initialization under distillation and fine-tuning settings. We also evaluate NeUQI combined with the Hadamard transform, a representative instance of the transformation-based techniques exemplified by QuIP, DuQuant, and FrameQuant, under both weight-only and weight-activation quantization. Following the previous work, all the quantized models are evaluated by measuring perplexity on the WikiText2 (**Wiki2**) (Merity et al., 2017) and **C4** (Raffel et al., 2020) validation sets, and zero-shot accuracy on five benchmarks: ARC-easy (**ArcE**), ARC-challenge (**ArcC**) (Clark et al., 2018), **PiQA** (Bisk et al., 2020), HellaSwag (**HellaS**) (Zellers et al., 2019), and WinoGrande (**WinoG**) (Sakaguchi et al., 2021).

## 5.2 IMPLEMENTATION SETTINGS

We evaluate weight-only quantization using channel- and group-wise schemes at 2, 3, and 4 bits, with the group size fixed to 128 as in prior work. For weight–activation quantization, we use channel-wise weights at 2 and 4 bits, with activations quantized to 4 bits using token-wise dynamic Min–Max initialization. We denote a configuration as WxAy, where x and y are the bit widths of weights and activations, respectively. Across all experiments, we employ GPTQ with column quantization ordered by decreasing activation L2-norm as the unified weight transformation method, ensuring fair comparison across different approaches. We follow GPTQ for calibration, using 128 C4 samples with 2048 tokens for LLaMA 2 and 4096 tokens for LLaMA 3 and Qwen 2.5. We perform distillation with 256 C4 samples for one epoch. We set NeUQI candidate space hyperparameters to $T = 2048$ and $T_c = 64$. For fairness, MagR is evaluated without extra coordinate descent iterations used in the original paper. More implementation details are included in Appendix D.

## 5.3 MAIN RESULTS

Under the most challenging 2-bit channel-wise quantization setting, our NeUQI achieves significant improvements over GPTQ, GPTAQ, MagR and OmniQuant, as shown in Table 1. Although the recent MagR method delivers acceptable performance on the Qwen 2.5 family, especially on perplexity, it performs poorly on the LLaMA family. In contrast, NeUQI consistently demonstrates strong performance across different model architectures and sizes. Similar conclusions hold under the 2-bit group-wise quantization setting, with detailed results shown in Table 8.

Moreover, as the bit-width increases, the performance gap between different methods gradually narrows, as shown in Table 9, Table 10, and Table 11 in the appendix. Our NeUQI continues to show stable advantages under the 3-bit setting, maintaining robustness across architectures. As for 4-bit, where all methods closely match the original non-quantized model, further improvements are limited. Nevertheless, NeUQI remains competitive and exhibits slight advantages in certain cases.

**Hadamard Transform** We further evaluate NeUQI in combination with the representative transformation-based technique, the Hadamard transform, under the weight-only setting W2A16 as well as the weight–activation settings W2A4 and W4A4. These experiments demonstrate that NeUQI can be effectively integrated with transformation-based techniques to further enhance performance. As shown in Table 2, NeUQI consistently outperforms GPTQ, GPTAQ, and MagR in these settings.

Table 3: Results of different initialization methods on LLaMA 2 and 3 family with 2-bit channel-wise quantization. † marks results from the original paper.

| Model | Size | Method | Wiki2↓ | C4↓ | ArcC↑ | ArcE↑ | HellaS↑ | PiQA↑ | WinoG↑ | Acc↑ |
|---|---|---|---|---|---|---|---|---|---|---|
| LLaMA 2 | 7B | Min-Max+ | 498 | 136 | 21.67 | 27.36 | 26.92 | 54.24 | 50.83 | 36.20 |
| | | MSE | 65.42 | 93.58 | 22.95 | 44.65 | 34.65 | 62.79 | 55.88 | 44.19 |
| | | LeanQuant† | 25.69 | 27.11 | 20.99 | 41.08 | 31.94 | 61.64 | 56.51 | 42.43 |
| | | Int-Search | 26.26 | 24.15 | 22.70 | 45.50 | 33.06 | 62.51 | 55.49 | 43.85 |
| | | NeUQI | **17.14** | **17.50** | **23.98** | **51.73** | **36.04** | **65.89** | **58.56** | **47.24** |
| | 13B | Min-Max+ | 32.19 | 25.96 | 22.61 | 32.03 | 32.97 | 57.40 | 50.83 | 39.17 |
| | | MSE | 155 | 223 | 19.88 | 28.41 | 29.04 | 55.44 | 49.33 | 36.42 |
| | | LeanQuant† | 24.43 | 20.92 | 24.32 | 50.88 | **38.01** | **67.19** | 56.91 | 47.46 |
| | | Int-Search | 15.67 | 16.69 | 26.11 | 53.32 | 36.48 | 65.51 | 56.59 | 47.60 |
| | | NeUQI | **13.72** | **14.39** | **26.19** | **55.18** | 37.37 | 65.94 | **59.12** | **48.76** |
| | 70B | Min-Max+ | 13.33 | 14.08 | 25.94 | 45.62 | 38.30 | 65.56 | 59.59 | 47.00 |
| | | MSE | 11.98 | 13.16 | 36.43 | 67.76 | 46.24 | 72.91 | 67.17 | 58.10 |
| | | LeanQuant† | 7.92 | 10.84 | - | - | - | - | - | - |
| | | Int-Search | 8.71 | 10.49 | 39.85 | 72.18 | 50.82 | 75.84 | 69.38 | 61.61 |
| | | NeUQI | **7.03** | **8.88** | **44.71** | **75.97** | **53.62** | **77.64** | **73.72** | **65.13** |
| LLaMA 3 | 8B | Min-Max+ | 3016 | 716 | 20.99 | 25.04 | 26.28 | 53.16 | 50.43 | 35.18 |
| | | MSE | 147 | 57.94 | 18.52 | 32.11 | 33.85 | 56.31 | 48.30 | 37.82 |
| | | LeanQuant† | - | - | 18.26 | 35.06 | 31.43 | 59.30 | 51.85 | 39.18 |
| | | Int-Search | 93.64 | 47.76 | 18.86 | 37.84 | 32.62 | 60.28 | 54.30 | 40.78 |
| | | NeUQI | **64.47** | **39.41** | **24.83** | **52.10** | **37.50** | **63.00** | **59.04** | **47.30** |

Table 4: Performance comparison on LLaMA 2 7B under 2-bit group-wise quantization with approximately equal average bit-width.

| Group | Avg Bits | Method | Wiki2↓ | C4↓ | ArcC↑ | ArcE↑ | HellaS↑ | PiQA↑ | WinoG↑ | Acc↑ |
|---|---|---|---|---|---|---|---|---|---|---|
| 64 | 2.28 | GPTQ | 16.03 | 16.04 | 22.87 | 45.79 | 39.43 | 65.56 | 57.06 | 46.14 |
| 64 | 2.28 | GPTAQ | 11.88 | 12.56 | 25.68 | 53.45 | 42.27 | 68.5 | 58.72 | 49.73 |
| 64 | 2.28 | MagR | 14.58 | 15.44 | 25.17 | 52.86 | 37.42 | 67.57 | 58.48 | 48.30 |
| 64 | 2.28 | Int-Search | 14.67 | 15.96 | 24.23 | 56.69 | 37.8 | 67.46 | 60.69 | 49.38 |
| 128 | 2.25 | NeUQI | 12.35 | 13.48 | 25.26 | 59.72 | 40.71 | 69.64 | 61.33 | 51.33 |
| 64 | 2.5 | NeUQI | 10.61 | 12.03 | 27.99 | 61.45 | 42.55 | 70.84 | 63.3 | 53.22 |

## 5.4 BIT ANALYSIS

**Integer Constraint**   We conduct an ablation study on the effects of initialization and the integer zero-point constraint, and compare our approach with several improved initialization methods that preserve the integer zero-point constraint, including Min-Max+[1] and MSE[2], as well as optimization-based LeanQuant and Int-Search, the latter sharing NeUQI's loss function 5. The differences between LeanQuant and Int-Search are discussed in Appendix D.4. As shown in Table 3, these methods improve upon Table 1 but still fall short of NeUQI. In the 2-bit setting, NeUQI gains superior results with less than 0.01 increase in average bit-width. This suggests that the common integer zero-point assumption is overly restrictive for optimization-based quantization and deserves further study.

**Bit Fairness**   For channel-wise quantization, removing the integer constraint leads to only a negligible increase in average bit-width. In 2-bit quantization with group size 128, the average bit-width is about 2.14 with the constraint and 2.25 without it. To ensure fair comparison, we also evaluate methods with the integer constraint at group size 64, which yields an average bit-width of about 2.28. As shown in Table 4, NeUQI delivers superior performance even at a lower average bit-width. Furthermore, double quantization (Dettmers et al., 2023) can further reduce the memory footprint of quantization parameters, mitigating the overhead of removing the integer constraint.

---

[1]A variant derived from correcting the flawed intuition of Min–Max, as detailed in Appendix A.
[2]A legacy MSE-based grid search method implemented in the GPTQ repository.

Table 5: Results of distillation on LLaMA 2 7B with 2-bit quantization and group size 128. † marks results from Li & Panda (2024), ‡ marks group size 64, * marks lightweight distillation, and **Tokens** indicates training set size times epochs.

| Method | Tokens | Wiki2↓ | C4↓ | ArcC↑ | ArcE↑ | HellaS↑ | PiQA↑ | WinoG↑ | Acc↑ |
|---|---|---|---|---|---|---|---|---|---|
| GPTQ* | $\sim 0.5M \times 1$ | 14.95 | 13.31 | 25.34 | 49.20 | 39.13 | 67.63 | 58.96 | 48.05 |
| OmniQuant† | $\sim 0.25M \times 20$ | 11.06 | 16.34 | 24.83 | 51.13 | 40.30 | 64.79 | 56.90 | 47.59 |
| OmniQuant†‡ | $\sim 0.25M \times 20$ | 9.62 | 13.79 | - | - | - | - | - | - |
| PV-tuning† | $\sim 1B \times 1$ | 8.49 | 10.78 | - | - | - | - | - | 52.17 |
| NeUQI* | $\sim 0.5M \times 1$ | **8.38** | **9.81** | **32.00** | **66.79** | **47.73** | **73.34** | **64.01** | **56.77** |

Table 6: Results of EfficientQAT and its NeUQI-initialized variant under 2-bit quantization with group size 128 on LLaMA 2 7B and LLaMA 3 8B. † marks results from the original paper.

| Model | Method | Wiki2↓ | C4↓ | ArcC↑ | ArcE↑ | HellaS↑ | PiQA↑ | WinoG↑ | Acc↑ |
|---|---|---|---|---|---|---|---|---|---|
| LLaMA 2 7B | EfficientQAT† | 7.19 | 8.79 | 36.52 | 69.78 | 50.84 | 74.16 | **66.22** | 59.50 |
| | +NeUQI | **6.81** | **8.41** | **37.88** | **69.82** | **52.14** | **75.57** | 64.88 | **60.01** |
| LLaMA 3 8B | EfficientQAT† | 9.80 | 13.22 | 36.01 | 69.15 | 50.74 | 75.30 | **65.67** | 59.37 |
| | +NeUQI | **9.14** | **12.51** | **38.48** | **71.93** | **51.96** | **76.61** | 64.40 | **60.68** |

## 5.5 DISCUSSIONS

**Distillation and Fine-tuning** As shown in Table 5, applying our lightweight distillation to GPTQ yields performance comparable to OmniQuant, while applying it to NeUQI surpasses both PV-tuning and OmniQuant. Notably, this is achieved with only about 0.5M tokens, whereas PV-tuning and OmniQuant require more memory, incur longer per-step runtimes than standard fine-tuning, and demand substantially more data. These results highlight the importance of better initialization, as poor initialization appears difficult to remedy even with heavy fine-tuning. Moreover, methods that go beyond the first specification can benefit from strong non-fine-tuning PTQ approaches such as NeUQI.

To further demonstrate the advantage of the better initialization within fine-tuning methods, we compare the very strong fine-tuning method EfficientQAT with a modified version that replaces only the quantization parameter initialization with NeUQI. The results shown in Table 6 indicate that even for such a strong fine-tuning method, NeUQI still provides further performance improvements.

**Comparison with Non-Quantized Models** With comparable memory usages, the quantized models using our NeUQI consistently outperform the non-quantized ones, as shown in Table 12. Although limited by computational resources and unable to experiment on larger models, these results demonstrate the potential of NeUQI-based low-bit-width quantization when applied to ultra-large models (e.g., DeepsSeek V3 671B (DeepSeek-AI et al., 2025)). They often lack smaller versions and remain difficult to deploy, particularly in resource-limited environments, even with 4-bit quantization.

## 6 CONCLUSION

To the best of our knowledge, we are the first to identify the limitations of the Min–Max strategic rationale. By moving beyond these limitations, NeUQI consistently outperforms existing methods and, with a reduced memory footprint, can even exceed the integer-constrained upper bound as well as the performance of the corresponding non-quantized model. Moreover, as an initialization method, NeUQI can also be combined with transformation-based techniques to further improve performance. The experiments on distillation and fine-tuning further validate its effectiveness and suggests that combining NeUQI with orthogonal techniques could yield additional gains. For future work, we plan to further explore the potential of uniform quantization by investigating fine-tuning strategies beyond the first specification to enhance post-initialization performance.

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

## A  MIN-MAX INTUITIVE FALLACY

Intuitively, Min-Max maps the entire observed range $[x_{\min}, x_{\max}]$ onto the low-error interval $[-zs, (-z + 2^k - 1)s]$. However, the more accurate and appropriate interval should be $[(-z - \frac{1}{2})s, (-z + 2^k - \frac{1}{2})s]$. Based on this correction, we simply propose an improved initialization strategy, **Min-Max+**, defining the quantization parameters as follows:

$$s = \frac{x_{max} - x_{min}}{2^k}, \quad z = -\left\lfloor \frac{x_{min}}{s} + \frac{1}{2} \right\rceil, \tag{11}$$

which matches the configuration that minimizes expected error when weights are ideally drawn independently and uniformly from an interval, as detailed with the theoretical analysis in Appendix B. Empirically, our simple adjustment leads to measurable gains, which confirms the flawed intuition of original Min-Max method. In real-world settings such as GPTQ quantization of the LLaMA 2 7B, Min-Max+ consistently outperforms Min-Max, as shown in Table 7.

Table 7: Comparison between Min–Max and Min–Max+ using GPTQ on the LLaMA-2-7B model. **Wx** denotes x-bit quantization and **Gy** denotes group size y.

| Setting | W3G128 | | W3 | | W2G128 | | W2 | |
|---|---|---|---|---|---|---|---|---|
| Method | Wiki2↓ | C4↓ | Wiki2↓ | C4↓ | Wiki2↓ | C4↓ | Wiki2↓ | C4↓ |
| Min-Max | 6.34 | 7.86 | 8.45 | 9.87 | 26.31 | 58.33 | 6953 | 2592 |
| Min-Max+ | **6.32** | **7.72** | **7.28** | **8.84** | **13.64** | **13.64** | **497** | **135** |

## B  OPTIMAL QUANTIZATION PARAMETER

**Lemma 1.** *If weights are independently and uniformly drawn from an interval $[a, b]$, then the optimal parameters to for $k$-bit uniform quantization to minimize excepted quantization error $\mathbb{E}_{x \sim \mathcal{U}(a,b)}[(Q_{s,z}(x) - x)^2]$ are given by*

$$s = \frac{b - a}{2^k}, \quad z = -\left( \frac{a}{s} + \frac{1}{2} \right). \tag{12}$$

*Proof.* We relax the uniform spacing constraint on $\mathcal{Q}$ to allow non-uniform intervals, but will show the minimum MSE occurs when they are all equal. The expected quantization error is

$$\mathcal{E} = \frac{1}{b-a} \sum_{i=0}^{N-1} \int_{b_i}^{b_{i+1}} (x - q_i)^2 \, dx, \tag{13}$$

where $N = 2^k$, $a = b_0 < \cdots < b_N = b$, and each $[b_i, b_{i+1})$ is mapped to $q_i$.

Let

$$\Delta_i = b_{i+1} - b_i. \tag{14}$$

For fixed $\Delta_i$, minimizing the integral

$$\int_{b_i}^{b_{i+1}} (x - q_i)^2 \, dx \tag{15}$$

yields

$$q_i^* = \frac{1}{\Delta_i} \int_{b_i}^{b_{i+1}} x \, dx = \frac{b_i + b_{i+1}}{2}, \tag{16}$$

and thus

$$\int_{b_i}^{b_{i+1}} (x - q_i^*)^2 \, dx = \frac{\Delta_i^3}{12}. \tag{17}$$

Substituting equation 17 into equation 13 gives

$$\mathcal{E} = \frac{1}{12(b-a)} \sum_{i=0}^{N-1} \Delta_i^3. \tag{18}$$

Since $f(\Delta) = \Delta^3$ is strictly convex on $(0, \infty)$, Jensen's inequality shows $\mathcal{E}$ is minimized when all $\Delta_i$ are equal:

$$\Delta_i = \frac{b-a}{N} = \frac{b-a}{2^k}. \tag{19}$$

Therefore the optimal uniform quantizer parameters are

$$s = \Delta_i = \frac{b-a}{2^k}, \quad z = -\frac{q_0^*}{s} = -\left( \frac{a}{s} + \frac{1}{2} \right). \tag{20}$$

$\square$

## C  ZERO-POINT ALGORITHMS

Detailed zero-point algorithms corresponding to Section 4.2.1 are provided below.

---

**Algorithm 2** Optimal Zero-point over Smoothed Objective

---

**Require:** Samples $\{x_i\}_{i=1}^n$, bit-width $k$
**Ensure:** Optimal zero-point $z_{\text{opt}}^{\text{smooth}}$ minimizing smoothed quantization loss
1: Initialize list of transition points: $\mathcal{T} \leftarrow []$
2: **for** each sample $x_i$ **do**
3:      Enter transition: $t_{\text{enter}} \leftarrow -\frac{1}{2} - x_i$
4:      Enter loss Function increment: $\delta_{\text{enter}}(z) \leftarrow h_i(x_i + z)^2 - h_i\frac{1}{4}$
5:      Exit transition: $t_{\text{exit}} \leftarrow 2^k - \frac{1}{2} - x_i$
6:      Exit loss Function increment: $\delta_{\text{exit}}(z) \leftarrow h_i\frac{1}{4} - h_i(x_i + z - (2^k - 1))^2$
7:      Add $(t_{\text{enter}}, \delta_{\text{enter}}(z))$ and $(t_{\text{exit}}, \delta_{\text{exit}}(z))$ to $\mathcal{T}$
8: **end for**
9: Sort $\mathcal{T}$ by transition point $t$
10: Get interval-wise loss function: $\mathcal{L}^{\text{I}}(z) \leftarrow \sum_i h_i(x_i + z - 0)^2$
11: **for** each $(t, \delta(z))$ in $\mathcal{T}$ **do**
12:      $\mathcal{L}^{\text{I}}(z) \leftarrow \mathcal{L}^{\text{I}}(z) + \delta(z)$
13:      Let next transition point be $t'$ (or $+\infty$ if none)
14:      Get interval minimum: $z' \leftarrow \arg\min_{z \in [t, t']} \mathcal{L}^{\text{I}}(z)$
15:      Evaluate $\mathcal{L}^{\text{I}}(z)$ at $z'$, update $z_{\text{opt}}^{\text{smooth}}$ if smaller
16: **end for**
17: **return** $z_{\text{opt}}^{\text{smooth}}$ with minimal loss

---

**Algorithm 3** Optimal Zero-point over Piecewise Quadratic Function in Limited Interval

---

**Require:** Samples $\{x_i\}_{i=1}^n$, bit-width $k$, interval $[z_l, z_r]$
**Ensure:** Optimal zero-point $z^*$ minimizing simplified quantization loss in interval $[z_l, z_r]$
1: Initialize list of transition points: $\mathcal{T} \leftarrow []$
2: **for** each sample $x_i$ **do**
3:      **for** $j = 0$ to $2^k - 2$ **do**
4:          Compute transition point: $t \leftarrow j + \frac{1}{2} - x_i$
5:          Compute loss Function increment: $\delta(z) \leftarrow h_i(x_i + z - (j+1))^2 - h_i(x_i + z - j)^2$
6:          **if** $t \in [z_l, z_r)$ **then**
7:             Add $(t, \delta(z))$ to $\mathcal{T}$
8:          **end if**
9:      **end for**
10: **end for**
11: Sort $\mathcal{T}$ by transition point $t$
12: Get interval-wise loss function: $\mathcal{L}^{\text{I}}(z) \leftarrow \sum_i h_i\big(x_i + z - \text{clip}(\lfloor x_i + z_l \rceil, 0, 2^k - 1)\big)^2$
13: Let $(t_{\text{first}}, \delta_{\text{first}}) \leftarrow$ the first element of $\mathcal{T}$
14: Get interval minimum: $z' \leftarrow \arg\min_{z \in [z_l, t_{\text{first}}]} \mathcal{L}^{\text{I}}(z)$
15: Evaluate $\mathcal{L}^{\text{I}}(z)$ at $z'$, record as current best: $z^* \leftarrow z'$
16: **for** each $(t, \delta)$ in $\mathcal{T}$ **do**
17:      $\mathcal{L}^{\text{I}}(z) \leftarrow \mathcal{L}^{\text{I}}(z) + \delta(z)$
18:      Let next transition point be $t'$ (or $+\infty$ if none)
19:      Get interval minimum: $z' \leftarrow \arg\min_{z \in [t, t']} \mathcal{L}^{\text{I}}(z)$
20:      Evaluate $\mathcal{L}^{\text{I}}(z)$ at $z'$, update $z^*$ if smaller
21: **end for**
22: **return** $z^*$ with minimal loss

---

# D IMPLEMENTATION DETAILS

## D.1 SETTING

### D.1.1 CALIBRATION STAGE

During the calibration stage, we follow the implementation of the official GPTQ repository,[3] and draw the same 128 samples from the C4 dataset for quantization calibration. For implementation convenience, we uniformly adopt `bfloat16` for all experiments. The token length of each sample is determined based on the characteristics and design intent of the target family. Especially for the LLaMA 2 family, each sample consists of 2048 tokens, consistent with the configurations adopted in GPTQ and LeanQuant. For the Qwen 2.5 models, each sample comprises 4096 tokens, as indicated in the Qwen 2.5 Technical Report, which specifies a 4096-token context length in its base configuration before context expansion. The same token length setting is applied to the LLaMA 3 family to ensure consistency. All experiments are conducted on NVIDIA A40 GPUs and NVIDIA L40 GPUs.

### D.1.2 DISTILLATION STAGE

In contrast to the calibration stage, the distillation utilizes 256 samples from the C4 dataset. The token lengths per sample remain consistent with those used during calibration. Furthermore, the computation precision remains `bfloat16`, identical to that in the calibration phase.

For the broader fine-tuning setup, we employ the AdamW optimizer with zero weight decay and momentum parameters set as $\beta = (0.9, 0.95)$. The learning rate is chosen via grid search over five candidate values: 1e-5, 3e-5, 1e-4, 3e-4, and 1e-3. We apply a cosine learning rate scheduler with a 10% warm-up ratio. In terms of hardware and infrastructure, we fine-tune LLaMA 2 7B in mixed precision using `DeepSpeed`[4], with batch size of 1 on a single NVIDIA A40 GPU.

### D.1.3 EVALUATION STAGE

Regarding perplexity evaluation, we follow the procedure implemented in the official GPTQ repository, specifically the `datautils.py`[5] and the `llama.py`[6] script. Perplexity scores are computed on the validation sets of the C4 and WikiText2 datasets, using the dataset identifiers `c4` and `wikitext2` as specified in the original implementation.

For zero-shot accuracy evaluation, we adopt the `lm-evaluation-harness`[7], a standardized and extensible framework for evaluating language models across diverse tasks. Inference is conducted using `vLLM` as the backend to ensure high throughput and memory efficiency. We evaluate five benchmark tasks: `ARC-Easy`, `ARC-Challenge`, `WinoGrande`, `HellaSwag`, and `PIQA`, whose corresponding identifiers in `lm-evaluation-harness` are `arc_easy`, `arc_challenge`, `winogrande`, `hellaswag`, and `piqa`, respectively.

## D.2 QUANTIZATION

We adopt the quantization strategy implemented in the official GPTQ repository. Specifically, we perform sequential quantization by quantizing layers in the order of the model forward pass. Each layer receives inputs from the already quantized preceding layers rather than from the original model, ensuring that the quantization is based on realistic activation distributions that more closely resemble those encountered during inference. Furthermore, we conduct the entire quantization process under full-precision settings to ensure numerical stability. For NeUQI, the grid search over the scale hyperparameter is conducted with $T = 2048$ and $T_c = 64$.

---

[3] https://github.com/IST-DASLab/gptq
[4] https://github.com/microsoft/DeepSpeed
[5] https://github.com/IST-DASLab/gptq/blob/main/datautils.py
[6] https://github.com/IST-DASLab/gptq/blob/main/llama.py
[7] https://github.com/EleutherAI/lm-evaluation-harness

## D.3 GPTQ

We implement the method using the LDLQ approach. The mathematical proof of their equivalence is in Appendix D.3.3. Moreover, LDLQ is computationally more efficient, as it requires fewer matrix operations and avoids the numerically unstable process of matrix inversion.

### D.3.1 GPTQ Implementation

GPTQ builds upon the principle of Optimal Brain Surgeon (OBS), designed for pruning. OBS applies a Taylor expansion of the loss function around the original weight $w$. Under the assumption that the model has reached convergence, the first-order term vanishes as a result of the stationarity condition, and higher-order terms of order three and above are considered negligible. Therefore, the dominant contribution is given by:

$$\mathcal{L}(\hat{\boldsymbol{w}}) \approx \frac{1}{2}(\hat{\boldsymbol{w}} - \boldsymbol{w})^\top \frac{\partial^2 \mathcal{L}}{\partial \hat{\boldsymbol{w}}^2}(\hat{\boldsymbol{w}} - \boldsymbol{w}) + \mathcal{L}(\boldsymbol{w}) \tag{21}$$

The quantization loss for a single-row in linear layer is the squared $\ell_2$-norm between the quantized output and the original output:

$$\mathcal{L}(\hat{w}) = \frac{1}{2}\|\boldsymbol{X}\hat{\boldsymbol{w}} - \boldsymbol{X}\boldsymbol{w}\|_2^2, \tag{22}$$

where $\hat{\boldsymbol{w}}$ denotes the quantized weight vector and $\boldsymbol{w}$ represents the original weight vector. This formulation follows a standard least-squares loss structure. The Hessian matrix of this loss with respect to $\hat{\boldsymbol{w}}$ is computed as:

$$\nabla_{\hat{\boldsymbol{w}}}^2 \mathcal{L}(\hat{\boldsymbol{w}}) = \frac{\partial^2}{\partial \hat{\boldsymbol{w}}^2} \frac{1}{2}\|\boldsymbol{X}\hat{\boldsymbol{w}} - \boldsymbol{X}\boldsymbol{w}\|_2^2 = \boldsymbol{X}^\top \boldsymbol{X}. \tag{23}$$

Thus, the matrix $\boldsymbol{H}$, representing the Hessian matrix, is given by:

$$\boldsymbol{H} = \nabla_{\hat{\boldsymbol{w}}}^2 \mathcal{L}(\hat{\boldsymbol{w}}) = \boldsymbol{X}^\top \boldsymbol{X}. \tag{24}$$

To ensure that the Hessian matrix $\boldsymbol{H}$ is positive definite, it is common practice to add a small scalar $\lambda > 0$ to its diagonal entries. This modification is equivalent to augmenting the quantization loss with a regularization term $\frac{\lambda}{2}\|\hat{\boldsymbol{w}} - \boldsymbol{w}\|^2$, which penalizes large deviations between the quantized weights $\hat{\boldsymbol{w}}$ and the original weights $\boldsymbol{w}$. In practice, $\lambda$ is set to $0.01$ times the mean of the diagonal entries of $H$, following GPTQ.

Given this formulation, we consider the problem of quantizing a single column vector $\boldsymbol{w}$ from the weight matrix. Specifically, let us focus on quantizing the element at a particular position $p$ within the vector $\boldsymbol{w}$. In other words, $w_p$ denotes the scalar entry at the $p$-th position of $\boldsymbol{w}$. We define a compensation vector $\delta^p$, which compensates for the quantization-induced error in $\boldsymbol{w}$ resulting from quantizing $w_p$. In this formulation, the update at coordinate $p$ reflects the quantization of $w_p$, while changes at other coordinates are to compensate for the loss caused by this quantization. The quantized vector is then given by $\hat{\boldsymbol{w}} = \boldsymbol{w} + \delta^p$, with the constraint that the $p$-th coordinate of $\hat{\boldsymbol{w}}$ matches the quantized value. Accordingly, we solve the following optimization problem:

$$\begin{aligned} \min_{\delta^p} \quad & (\delta^p)^\top \boldsymbol{H} \delta^p \\ \text{s.t.} \quad & e_p^\top \delta^p = \hat{w}_p - w_p \end{aligned} \tag{25}$$

Using the method of Lagrange multipliers, we construct the Lagrangian:

$$\mathcal{L}(\delta^p, \lambda) = (\delta^p)^\top \boldsymbol{H} \delta^p + \lambda \left( e_p^\top \delta^p - (\hat{w}_p - w_p) \right) \tag{26}$$

Taking the gradient with respect to $\delta^p$ and setting it to zero:

$$\nabla_{\delta^p} \mathcal{L} = 2\boldsymbol{H}\delta^p + \lambda e_p = 0 \quad \Rightarrow \quad \delta^p = -\frac{\lambda}{2}\boldsymbol{H}^{-1} e_p \tag{27}$$

Substituting back into the constraint:

$$e_p^\top \delta^p = -\frac{\lambda}{2} e_p^\top \boldsymbol{H}^{-1} e_p = \hat{w}_p - w_p \Rightarrow \lambda = -\frac{2(\hat{w}_p - w_p)}{(\boldsymbol{H}^{-1})_{p,p}} \tag{28}$$

Thus, we obtain:

$$\delta^p = \frac{(\hat{w}_p - w_p)}{(\boldsymbol{H}^{-1})_{p,p}} \boldsymbol{H}^{-1} e_p, \quad \mathcal{L}(\delta^p) = \frac{(\hat{w}_p - w_p)^2}{(\boldsymbol{H}^{-1})_{p,p}} \tag{29}$$

Based on this result, we can adopt a greedy strategy: at each iteration, the position $p$ that results in the minimal increase in quantization loss $\mathcal{L}(\delta^p)$ is selected, the corresponding update is applied. However, to better utilize GPU parallelism and improve computational efficiency, we follow a heuristic implemented in the GPTQ repository, specifically in the `gptq.py` script [8]. In this approach, the indices are sorted and traversed in decreasing order of the Hessian diagonals $H_{p,p}$.

Assuming we fix a quantization order $i = 1, 2, \ldots, n$, we define:

$$\delta_{i:}^i = \frac{(H_{i:,i:})^{-1} e_i}{((H_{i:,i:})^{-1})_{ii}}, \quad \delta_{:i}^i = \boldsymbol{0} \tag{30}$$

where, $\delta^i$ denotes the update basis vector for position $i$, satisfying $(\delta^i)_i = 1$. The actual update applied to the weight vector is $(\hat{w}_i - w_i)\delta^i$.

For any $j \geq i$, we have:

$$H_{j,:}\delta^i = e_j^\top H_{i:,i:} \frac{(H_{i:,i:})^{-1} e_i}{((H_{i:,i:})^{-1})_{ii}} = 1_{i=j} \left( ((H_{i:,i:})^{-1})_{ii} \right)^{-1} \tag{31}$$

Therefore, the update matrix satisfies:

$$\boldsymbol{H}\Delta = \boldsymbol{U}, \tag{32}$$

where $\Delta = [\delta^1, \delta^2, \ldots, \delta^n]$ is a unit lower-triangular matrix, and the resulting matrix $\boldsymbol{U}$ is upper-triangular because for $j > i$, $U_{j,i} = 0$.

This relation implies that $H$ can be factorized as

$$\boldsymbol{H} = \boldsymbol{U}\Delta^{-1}, \tag{33}$$

which corresponds to a UL decomposition of $\boldsymbol{H}$, where $\Delta^{-1}$ is unit lower-triangular. On the other hand, the Hessian matrix $\boldsymbol{H}$ is positive-definite and thus admits a unique LDL decomposition of the form

$$\boldsymbol{H} = (\boldsymbol{L}^\top \boldsymbol{D})\boldsymbol{L}, \tag{34}$$

where $\boldsymbol{L}$ is also a unit lower-triangular matrix and $\boldsymbol{D}$ is diagonal matrix.

Since both equation 33 and equation 34 represent decompositions of the same matrix $\boldsymbol{H}$ with a unit lower-triangular factor on the right, and the unit lower-triangular factor in such factorizations is unique for positive-definite matrices, we conclude that $\Delta = \boldsymbol{L}^{-1}$.

Finally, the algorithm is as follows:

### D.3.2 LDLQ IMPLEMENTATION

LDLQ is a newly proposed method introduced in QuIP, which has been empirically validated to be equivalent to GPTQ. It provides an alternative understanding of the GPTQ algorithm through a structurally distinct formulation. Specifically, when quantizing a single row of the weight matrix, LDLQ proceeds coordinate-wise in a fixed order: each coordinate is quantized one at a time, and the unquantized coordinates are adjusted to compensate for the quantization error. This process is captured through a strictly lower-triangular compensation matrix $L$, which defines the final quantized weights via

$$\hat{\boldsymbol{w}} = Q(\tilde{\boldsymbol{w}}), \quad \tilde{\boldsymbol{w}} = \boldsymbol{w} - L(\hat{\boldsymbol{w}} - \boldsymbol{w}) \tag{35}$$

---

[8] https://github.com/IST-DASLab/gptq/blob/main/gptq.py

---

**Algorithm 4** GPTQ Quantization

---

**Require:** Hessian matrix $\boldsymbol{H}$, original weight vector $\boldsymbol{w}$, quantization operator $Q(\cdot)$
1: Compute LDL decomposition: $\boldsymbol{H} = \boldsymbol{L}^\top \boldsymbol{D} \boldsymbol{L}$
2: Compute inverse of unit lower-triangular matrix: $\Delta = \boldsymbol{L}^{-1}$
3: Compute compensation matrix: $\boldsymbol{L} \leftarrow \Delta - I$
4: Initialize quantized weight $\hat{w}$ with zeros
5: **for** each index $i = 1$ to $n$ **do**
6: $\quad \hat{w}_i \leftarrow Q(w_i)$
7: $\quad \boldsymbol{w} \leftarrow \boldsymbol{w} + L_{:,i} \cdot (\hat{w}_i - w_i)$
8: **end for**
9: **return** $\hat{w}$

---

The quantization-induced error is

$$\eta = \hat{\boldsymbol{w}} - \tilde{\boldsymbol{w}} = (\boldsymbol{I} + \boldsymbol{L})(\hat{\boldsymbol{w}} - \boldsymbol{w}) \tag{36}$$

Solving for $\hat{\boldsymbol{w}} - \boldsymbol{w}$ yields

$$\hat{\boldsymbol{w}} - \boldsymbol{w} = (\boldsymbol{I} + \boldsymbol{L})^{-1}\eta \tag{37}$$

Substituting into the original quadratic loss function gives

$$\mathcal{L}(\hat{\boldsymbol{w}}) = \eta^\top (\boldsymbol{I} + \boldsymbol{L})^{-\top} \boldsymbol{H} (\boldsymbol{I} + \boldsymbol{L})^{-1}\eta \tag{38}$$

Given the LDL decomposition $\boldsymbol{H} = \boldsymbol{L'}^\top \boldsymbol{D} \boldsymbol{L'}$, LDLQ claims that the minimal loss is achieved when $\boldsymbol{I} + \boldsymbol{L} = \boldsymbol{L'}$, resulting in

$$\mathcal{L}(\hat{\boldsymbol{w}}) = \eta^\top \boldsymbol{D} \eta \tag{39}$$

This diagonalization simplifies the loss into a weighted squared norm of the quantization error, enabling an efficient sequential update procedure as described below:

---

**Algorithm 5** LDLQ Quantization

---

**Require:** Hessian matrix $\boldsymbol{H}$, original weight vector $\boldsymbol{w}$, quantization operator $Q(\cdot)$
1: Compute LDL decomposition: $\boldsymbol{H} = \boldsymbol{L'}^\top \boldsymbol{D} \boldsymbol{L'}$
2: Initialize quantized weight vector $\hat{w}$ with zeros
3: Compute compensation matrix $\boldsymbol{L} \leftarrow \boldsymbol{L'} - \boldsymbol{I}$
4: **for** each index $i = 1$ to $n$ **do**
5: $\quad \tilde{w}_i \leftarrow w_i - L_{i,:} \cdot (\hat{w}_i - w_i)$
6: $\quad \hat{w}_i \leftarrow Q(\tilde{w}_i)$
7: **end for**
8: **return** $\hat{w}$

---

### D.3.3 EQUIVALENCE BETWEEN GPTQ AND LDLQ

The following interpretation reinforces the equivalence between GPTQ and LDLQ. In particular, we can rewrite the GPTQ formulation as follows:

$$\hat{\boldsymbol{w}} = Q(\tilde{\boldsymbol{w}}), \quad \tilde{\boldsymbol{w}} = \boldsymbol{w} + (\Delta - \boldsymbol{I})(\hat{\boldsymbol{w}} - \tilde{\boldsymbol{w}}) \tag{40}$$

We can derive a more explicit form of $\tilde{\boldsymbol{w}}$ by manipulating the second equation in equation 40 as follows:

$$\tilde{\boldsymbol{w}} = \boldsymbol{w} + (\Delta - \boldsymbol{I})(\hat{\boldsymbol{w}} - \tilde{\boldsymbol{w}}) \tag{41}$$
$$\Delta \tilde{\boldsymbol{w}} = \boldsymbol{w} + (\Delta - \boldsymbol{I})\hat{\boldsymbol{w}} \tag{42}$$
$$\tilde{\boldsymbol{w}} = \Delta^{-1}\boldsymbol{w} + (\boldsymbol{I} - \Delta^{-1})\hat{\boldsymbol{w}} \tag{43}$$
$$\tilde{\boldsymbol{w}} = \boldsymbol{w} - (\Delta^{-1} - \boldsymbol{I})(\hat{\boldsymbol{w}} - \boldsymbol{w}) \tag{44}$$

On the other hand, the LDLQ formulation is given by:

$$\tilde{w} = w - (L' - I)(\hat{w} - w) \tag{45}$$

By comparing equations equation 44 and equation 45, we observe that the two formulations become equivalent if $\Delta^{-1} = L'$. As established in the previous two sections, both $\Delta^{-1}$ and $L'$ correspond to the $L$ factor derived from the LDL decomposition. Therefore, LDLQ is mathematically equivalent to GPTQ.

### D.4 DIFFERENCE BETWEEN LEANQUANT AND INT-SEARCH

LeanQuant$_{aff}$, short for LeanQuant, and Int-Search adopt the same loss formulation of the form

$$\mathcal{L}(s, z) = \sum_i h_i (Q_{s,z}(w_i) - w_i)^2, \tag{46}$$

but differ fundamentally in how the weight importance $h_i$ is defined and how to find the optimal scale and zero-point.

**Weight importance**    In LeanQuant, the weight importance $h_i$ is inspired by the objective used in the first step of GPTQ, which selects the optimal quantization point by minimizing

$$\frac{(Q_{s,z}(w_i) - w_i)^2}{(H^{-1})_{ii}}, \tag{47}$$

leading to $h_i = ((H^{-1})_{ii})^{-1}$, and further generalized to $h_i = ((H^{-1})_{ii})^{-p}$ with a tunable strength parameter $p$. In contrast, Int-Search adopts the diagonal loss approximation by directly setting $h_i = H_{ii}$, avoiding the need to invert the Hessian matrix.

**Parameter Search**    LeanQuant still operates within the Min-Max strategic rationale: its grid search essentially enumerates different min and max values, from which they derive the scale and zero-point using the standard Min-Max formula, with a time complexity of $\mathcal{O}(T^2 n)$, where $T$ (typically 2048) denotes the number of grid points and $n$ the number of weights. In contrast, Int-Search employs a grid search over candidate scale values, with the zero-point constrained to be a $k$-bit unsigned integer. A simple implementation of this scale-based search has a time complexity of $\mathcal{O}(T2^k n)$. To further accelerate initialization, we identify a more efficient alternative by switching the search perspective from scale to zero-point. For a fixed zero-point, the objective remains a piecewise quadratic function of the scale, similar to the case discussed in Section 4.2, which enables an optimized search using a similar method with total complexity reduced to $\mathcal{O}(2^k \cdot n2^k \log(n2^k))$, which is independent of $T$.

## E    HARDWARE SUPPORT FOR FLOATING-POINT ZERO-POINTS

For hardware support, the BitBLAS library[9], which has been integrated into the widely used inference framework vLLM[10], provides compatibility with mainstream GPUs such as A100 and RTX 4090, and supports zero-points represented in floating-point format when the parameter `zeros_mode` is set to `original`, as specified in the BitBLAS Python API[11]. This indicates that our proposed NeUQI has been supported by GPU kernels implemented in BitBLAS, without requiring specialized hardware modifications.

## F    LLM USAGE

We employ LLMs to polish the paper while preserving its original meaning. Specifically, LLMs are used to insert appropriate logical connectors for improved coherence, to select more academic and precise vocabulary for accurate expression, and to streamline overly redundant passages without altering their original meaning.

---

[9] https://github.com/microsoft/BitBLAS
[10] https://github.com/vllm-project/vllm
[11] https://github.com/microsoft/BitBLAS/blob/main/docs/PythonAPI.md

# G  Supplementary Results

This appendix provides supplementary quantization results. Table 8 presents detailed results of 2-bit quantization with a group size of 128 for LLaMA and Qwen families. Tables 9, 10, and 11 report detailed results for 3-bit and 4-bit quantization of LLaMA 2, LLaMA 3, and Qwen 2.5 families, respectively. Table 12 shows a comparison between quantized and non-quantized models from Qwen 2.5 family under comparable main body memory usage.

Table 8: Results of 2-bit group-wise quantization with group size of 128 on the LLaMA 2, LLaMA 3, and Qwen 2.5 families. † denotes that the calibration set used differs from the original paper.

| Model | Size | Bits | Method | Wiki2↓ | C4↓ | ArcC↑ | ArcE↑ | HellaS↑ | PiQA↑ | WinoG↑ | Acc↑ |
|---|---|---|---|---|---|---|---|---|---|---|---|
| LLaMA 2 | 7B | 2 | GPTQ | 26.31 | 23.52 | 22.70 | 36.91 | 34.39 | 60.50 | 54.62 | 41.82 |
| | | | GPTAQ | 15.80 | 15.11 | 25.85 | 48.44 | 38.45 | 66.97 | 58.41 | 47.63 |
| | | | MagR† | 15.41 | 15.59 | 24.66 | 47.90 | 39.19 | 65.89 | 58.80 | 47.29 |
| | | | NeUQI | **12.35** | **13.48** | **25.26** | **59.72** | **40.71** | **69.64** | **61.33** | **51.33** |
| | 13B | 2 | GPTQ | 12.50 | 13.16 | 26.54 | 53.58 | 42.00 | 68.34 | 55.01 | 49.09 |
| | | | GPTAQ | 10.35 | 11.22 | 29.95 | 58.25 | 45.24 | 70.51 | 62.19 | 53.23 |
| | | | MagR† | 17.15 | 11.35 | 30.46 | 61.74 | 45.72 | 71.60 | 60.62 | 54.03 |
| | | | NeUQI | **8.82** | **10.60** | **35.32** | **67.72** | **45.98** | **73.12** | **68.59** | **58.15** |
| | 70B | 2 | GPTQ | 7.04 | 8.62 | 38.48 | 70.20 | 53.00 | 75.14 | 69.93 | 61.35 |
| | | | GPTAQ | 6.77 | 8.27 | 36.69 | 67.85 | 52.13 | 74.59 | 70.96 | 60.44 |
| | | | MagR† | 10.87 | 9.21 | 41.89 | 68.64 | 55.13 | 68.50 | 73.64 | 61.56 |
| | | | NeUQI | **5.60** | **7.51** | **47.53** | **77.86** | **56.90** | **78.24** | **75.53** | **67.21** |
| LLaMA 3 | 8B | 2 | GPTQ | 210 | 121 | 19.54 | 28.91 | 26.94 | 54.73 | 51.62 | 36.35 |
| | | | GPTAQ | 81.25 | 51.94 | 18.94 | 34.93 | 31.19 | 59.14 | 50.28 | 38.90 |
| | | | MagR | 61.71 | 74.78 | 18.43 | 33.80 | 28.72 | 56.69 | 51.70 | 37.87 |
| | | | NeUQI | **28.90** | **26.83** | **23.29** | **50.04** | **36.04** | **63.38** | **59.91** | **46.53** |
| | 70B | 2 | GPTQ | 26.25 | 28.59 | 22.27 | 38.80 | 34.61 | 61.04 | 55.41 | 42.43 |
| | | | GPTAQ | 19.67 | 20.20 | 20.39 | 38.13 | 40.77 | 60.45 | 59.75 | 43.90 |
| | | | MagR | 30.86 | 55.42 | 26.71 | 51.56 | 40.78 | 67.74 | 64.56 | 50.27 |
| | | | NeUQI | **10.75** | **14.40** | **39.68** | **71.30** | **48.66** | **74.05** | **70.56** | **60.85** |
| Qwen 2.5 | 7B | 2 | GPTQ | 21.66 | 22.77 | 23.72 | 43.43 | 39.32 | 62.02 | 53.75 | 44.45 |
| | | | GPTAQ | 19.06 | 20.96 | 27.3 | 46.63 | 41.69 | 66.92 | 56.59 | 47.83 |
| | | | MagR | **13.65** | **16.56** | 30.46 | 61.95 | 43.88 | 68.50 | 60.30 | 53.02 |
| | | | NeUQI | 14.43 | 16.65 | **36.26** | **67.76** | **45.56** | **71.55** | **65.04** | **57.23** |
| | 14B | 2 | GPTQ | 18.62 | 19.22 | 22.95 | 42.80 | 38.38 | 61.97 | 51.62 | 43.54 |
| | | | GPTAQ | 15.89 | 16.7 | 25.6 | 51.98 | 42.98 | 67.79 | 58.09 | 49.29 |
| | | | MagR | 12.94 | 14.09 | 28.33 | 56.57 | 44.52 | 70.40 | 60.77 | 52.12 |
| | | | NeUQI | **10.57** | **13.61** | **43.52** | **75.63** | **50.26** | **74.92** | **72.30** | **63.32** |
| | 32B | 2 | GPTQ | 10.95 | 13.36 | 30.38 | 59.30 | 47.56 | 71.38 | 56.59 | 53.04 |
| | | | GPTAQ | 11.15 | 13.2 | 32.51 | 64.14 | 48.71 | 74.32 | 59.12 | 55.76 |
| | | | MagR | 8.82 | 11.65 | 36.69 | 69.57 | 52.23 | 75.57 | 64.40 | 59.69 |
| | | | NeUQI | **8.51** | **11.41** | **46.08** | **76.39** | **55.86** | **78.02** | **75.14** | **66.30** |
| | 72B | 2 | GPTQ | 10.35 | 12.27 | 35.49 | 63.51 | 50.11 | 72.36 | 59.35 | 56.17 |
| | | | GPTAQ | 11.61 | 11.98 | 37.8 | 67.47 | 53.33 | 74.76 | 65.04 | 59.68 |
| | | | MagR | 11.00 | 11.22 | 44.37 | 75.88 | 56.58 | 77.31 | 71.59 | 65.15 |
| | | | NeUQI | **6.63** | **9.97** | **51.88** | **81.61** | **59.23** | **79.87** | **75.69** | **69.66** |

Table 9: Results are reported for the LLaMA 2 family models under bfloat16, as well as for models quantized to 3-bit, 3-bit with group size 128, 4-bit, and 4-bit with group size 128. † denotes that we use bfloat16 for easier fine-tuning, whereas previous work uses float16.

| Size | Bits | Group | Method | Wiki2↓ | C4↓ | ArcC↑ | ArcE↑ | HellaS↑ | PiQA↑ | WinoG↑ | Acc↑ |
|------|------|-------|--------|--------|-----|-------|-------|---------|-------|--------|------|
| 7B | BF16† | - | - | 5.12 | 6.63 | 43.34 | 76.26 | 57.17 | 77.97 | 68.98 | 64.74 |
| | 4 | 128 | GPTQ | 5.62 | 7.12 | **43.52** | **75.55** | **56.27** | 77.48 | **69.85** | **64.53** |
| | | | NeUQI | **5.60** | **7.09** | 42.92 | 75.17 | 56.19 | **77.64** | 69.46 | 64.27 |
| | | - | GPTQ | 5.84 | 7.36 | 41.64 | 74.16 | **55.84** | 77.69 | **69.85** | 63.84 |
| | | | NeUQI | **5.80** | **7.26** | **41.89** | **74.87** | 54.78 | 76.77 | 68.51 | 63.36 |
| | 3 | 128 | GPTQ | 6.34 | 7.86 | **39.76** | **73.74** | 53.91 | **77.31** | 67.48 | **62.44** |
| | | | NeUQI | **6.07** | **7.58** | 38.57 | 72.35 | **54.19** | 76.44 | 67.48 | 61.81 |
| | | - | GPTQ | 8.45 | 9.87 | 34.73 | 66.46 | 49.08 | 73.34 | 64.40 | 57.60 |
| | | | NeUQI | **6.56** | **8.10** | **37.97** | **71.42** | **50.75** | **75.41** | **68.19** | **60.75** |
| 13B | BF16† | - | - | 4.57 | 6.05 | 48.21 | 79.46 | 60.09 | 79.11 | 72.30 | 67.83 |
| | 4 | 128 | GPTQ | 5.00 | **6.56** | **47.70** | 78.28 | **59.61** | 78.62 | **72.53** | 67.35 |
| | | | NeUQI | **4.98** | **6.56** | 47.10 | **79.21** | 59.55 | **79.11** | 72.30 | **67.45** |
| | | - | GPTQ | 5.15 | 6.71 | 44.97 | 77.44 | **58.88** | 77.91 | **71.27** | 66.09 |
| | | | NeUQI | **5.09** | **6.67** | **45.31** | **77.86** | 58.64 | **78.13** | 71.19 | **66.23** |
| | 3 | 128 | GPTQ | 5.43 | 7.05 | 45.22 | 77.23 | **58.14** | 77.69 | 70.72 | 65.80 |
| | | | NeUQI | **5.32** | **6.91** | **45.82** | **78.32** | 57.79 | **78.40** | **72.77** | **66.62** |
| | | - | GPTQ | 6.46 | 8.03 | 38.91 | 73.48 | 55.18 | 76.39 | 68.35 | 62.46 |
| | | | NeUQI | **5.70** | **7.25** | **42.75** | **75.84** | **56.06** | **77.64** | **70.72** | **64.60** |
| 70B | BF16† | - | - | 3.12 | 4.97 | 54.52 | 82.66 | 64.76 | 82.15 | 77.43 | 72.30 |
| | 4 | 128 | GPTQ | 3.42 | **5.58** | 54.69 | 82.74 | **64.45** | 81.83 | 77.19 | 72.18 |
| | | | NeUQI | **3.41** | **5.58** | **54.95** | **82.83** | 64.29 | **82.05** | **77.98** | **72.42** |
| | | - | GPTQ | 3.59 | 5.68 | 54.27 | 81.99 | **64.16** | **82.15** | 77.19 | 71.95 |
| | | | NeUQI | **3.47** | **5.62** | **54.35** | **83.08** | 64.04 | 81.66 | **78.53** | **72.33** |
| | 3 | 128 | GPTQ | 3.87 | 5.86 | 53.24 | 81.57 | **63.16** | 81.61 | **77.43** | 71.40 |
| | | | NeUQI | **3.71** | **5.77** | **54.95** | **82.45** | 62.99 | **81.99** | 77.03 | **71.88** |
| | | - | GPTQ | 4.83 | 6.57 | 49.15 | 79.38 | 60.60 | 80.36 | 74.35 | 68.77 |
| | | | NeUQI | **3.90** | **5.90** | **53.50** | **83.00** | **62.58** | **81.61** | **76.72** | **71.48** |

Table 10: Results are reported for the LLaMA 3 family models under bfloat16, as well as for models quantized to 3-bit, 3-bit with group size 128, 4-bit, and 4-bit with group size 128.

| Size | Bits | Group | Method | Wiki2↓ | C4↓ | ArcC↑ | ArcE↑ | HellaS↑ | PiQA↑ | WinoG↑ | Acc↑ |
|------|------|-------|--------|--------|-----|-------|-------|---------|-------|--------|------|
| 8B | BF16 | - | - | 5.76 | 8.32 | 50.34 | 80.22 | 60.19 | 79.60 | 73.64 | 68.80 |
| | 4 | 128 | GPTQ | 6.19 | 8.99 | 47.61 | 77.86 | 59.06 | 77.75 | 73.88 | 67.23 |
| | | | NeUQI | **6.12** | **8.87** | **50.00** | **80.18** | **59.53** | **78.67** | **74.90** | **68.66** |
| | | - | GPTQ | 6.97 | 9.95 | 44.54 | 77.27 | 57.87 | 77.15 | 73.32 | 66.03 |
| | | | NeUQI | **6.67** | **9.41** | **47.10** | **77.69** | **58.13** | **79.54** | **74.59** | **67.41** |
| | 3 | 128 | GPTQ | 8.30 | 11.50 | 40.19 | 71.84 | 54.43 | 76.22 | 70.72 | 62.68 |
| | | | NeUQI | **7.45** | **10.49** | **46.50** | **79.34** | **55.28** | **77.69** | **73.48** | **66.46** |
| | | - | GPTQ | 19.03 | 29.26 | 25.34 | 46.25 | 43.02 | 62.84 | 59.91 | 47.47 |
| | | | NeUQI | **9.70** | **11.61** | **43.77** | **76.30** | **53.87** | **77.20** | **71.35** | **64.50** |
| 70B | BF16 | - | - | 2.68 | 5.88 | 60.49 | 86.95 | 66.37 | 82.48 | 80.90 | 75.44 |
| | 4 | 128 | GPTQ | 3.40 | 6.41 | 57.76 | **85.14** | **66.00** | 82.05 | 80.03 | 74.20 |
| | | | NeUQI | **3.17** | **6.25** | **58.02** | 84.76 | 65.82 | **82.43** | **80.90** | **74.39** |
| | | - | GPTQ | 1486 | 1404 | 19.97 | 25.21 | 30.47 | 52.94 | 50.83 | 35.88 |
| | | | NeUQI | **4.90** | **10.00** | **51.96** | **80.47** | **62.29** | **79.49** | **63.61** | **67.57** |
| | 3 | 128 | GPTQ | 5.30 | 8.33 | 51.37 | 80.93 | **62.73** | **80.90** | 75.53 | 70.29 |
| | | | NeUQI | **4.63** | **7.58** | **54.52** | **83.42** | 62.50 | 80.47 | **80.03** | **72.19** |
| | | - | GPTQ | 2645 | 1111 | 20.82 | 25.04 | 26.16 | 52.18 | 51.22 | 35.08 |
| | | | NeUQI | **9.04** | **13.36** | **35.84** | **69.11** | **52.97** | **71.82** | **55.80** | **57.11** |

Table 11: Results are reported for the Qwen 2.5 family models under bfloat16, as well as for models quantized to 3-bit, 3-bit with group size 128, 4-bit, and 4-bit with group size 128.

| Size | Bits | Group | Method | Wiki2↓ | C4↓ | ArcC↑ | ArcE↑ | HellaS↑ | PiQA↑ | WinoG↑ | Acc↑ |
|------|------|-------|--------|--------|-----|-------|-------|---------|-------|--------|------|
| 7B | BF16 | - | - | 6.39 | 10.02 | 48.29 | 80.56 | 60.00 | 78.67 | 72.69 | 68.04 |
| | 4 | 128 | GPTQ | 6.61 | 10.19 | 48.21 | 80.35 | 59.24 | **79.05** | 71.35 | 67.64 |
| | | | NeUQI | **6.55** | **10.17** | **48.63** | **80.98** | **59.36** | 78.67 | **73.95** | **68.32** |
| | | - | GPTQ | 7.06 | 10.61 | 46.59 | 79.25 | 58.08 | 78.56 | 68.59 | 66.21 |
| | | | NeUQI | **6.82** | **10.42** | **48.55** | **79.38** | **58.63** | **78.78** | **71.90** | **67.45** |
| | 3 | 128 | GPTQ | 7.42 | 10.94 | 43.77 | 75.63 | **56.71** | 77.37 | 65.75 | 63.85 |
| | | | NeUQI | **7.14** | **10.80** | **46.42** | **80.05** | 56.46 | **78.35** | **70.40** | **66.34** |
| | | - | GPTQ | 10.73 | 13.90 | 39.33 | 71.21 | 50.16 | 73.29 | 60.77 | 58.95 |
| | | | NeUQI | **8.64** | **11.72** | **44.80** | **76.77** | **55.15** | **77.97** | **70.96** | **65.13** |
| 14B | BF16 | - | - | 4.93 | 8.75 | 55.80 | 82.37 | 63.38 | 81.07 | 74.66 | 71.46 |
| | 4 | 128 | GPTQ | 5.29 | 8.93 | 54.35 | 81.94 | **62.94** | 80.96 | 74.98 | 71.03 |
| | | | NeUQI | **5.20** | **8.89** | **56.40** | **83.25** | 62.78 | **81.01** | **75.61** | **71.81** |
| | | - | GPTQ | 5.80 | 9.24 | **52.73** | **81.44** | 62.27 | 80.20 | 74.98 | 70.32 |
| | | | NeUQI | **5.46** | **9.05** | 52.56 | 81.10 | **62.62** | **80.36** | **77.19** | **70.77** |
| | 3 | 128 | GPTQ | 6.29 | 9.63 | 48.55 | 79.38 | 60.06 | 79.16 | 72.14 | 67.86 |
| | | | NeUQI | **5.91** | **9.42** | **52.90** | **81.02** | **60.71** | **80.90** | **76.72** | **70.45** |
| | | - | GPTQ | 8.48 | 11.61 | 38.82 | 67.21 | 54.22 | 75.52 | 65.51 | 60.26 |
| | | | NeUQI | **6.82** | **9.89** | **50.17** | **80.43** | **59.39** | **79.54** | **76.09** | **69.12** |
| 32B | BF16 | - | - | 4.67 | 8.59 | 53.16 | 80.51 | 65.00 | 81.99 | 75.69 | 71.27 |
| | 4 | 128 | GPTQ | 4.87 | 8.69 | **54.10** | **81.44** | 64.59 | 80.69 | 75.93 | **71.35** |
| | | | NeUQI | **4.82** | **8.68** | 53.24 | 80.26 | **64.71** | **81.18** | **76.64** | 71.20 |
| | | - | GPTQ | 5.23 | 8.89 | 51.19 | 79.29 | 63.96 | 80.41 | 75.45 | 70.06 |
| | | | NeUQI | **4.98** | **8.77** | **53.07** | **81.48** | **64.46** | **81.77** | **77.35** | **71.63** |
| | 3 | 128 | GPTQ | 5.63 | 9.11 | 51.11 | 79.50 | 62.83 | 80.36 | 75.14 | 69.79 |
| | | | NeUQI | **5.30** | **8.99** | **51.71** | **82.07** | **62.92** | **80.90** | **76.40** | **70.80** |
| | | - | GPTQ | 7.21 | 10.36 | 44.03 | 74.96 | 58.75 | 77.97 | 66.22 | 64.38 |
| | | | NeUQI | **5.85** | **9.31** | **51.19** | **80.98** | **62.30** | **80.14** | **75.14** | **69.95** |
| 72B | BF16 | - | - | 3.64 | 7.75 | 58.11 | 84.76 | 67.59 | 82.10 | 77.35 | 73.98 |
| | 4 | 128 | GPTQ | 3.82 | 7.85 | **57.51** | **84.60** | 67.14 | **82.26** | 78.06 | 73.91 |
| | | | NeUQI | **3.78** | **7.83** | **57.51** | 84.47 | **67.16** | 81.99 | **78.93** | **74.01** |
| | | - | GPTQ | 4.17 | 8.35 | 55.38 | **84.68** | 66.68 | 81.66 | **77.90** | 73.26 |
| | | | NeUQI | **3.95** | **7.91** | **58.02** | 84.34 | **66.81** | **82.15** | **77.90** | **73.85** |
| | 3 | 128 | GPTQ | 4.55 | 8.53 | **55.97** | **83.92** | 65.66 | 80.79 | 77.74 | 72.82 |
| | | | NeUQI | **4.26** | **8.11** | 54.52 | 83.75 | **65.85** | **82.21** | **78.06** | **72.88** |
| | | - | GPTQ | 6.89 | 9.96 | 46.93 | 78.03 | 61.51 | 78.62 | 71.43 | 67.30 |
| | | | NeUQI | **4.99** | **8.37** | **56.40** | **83.75** | **64.92** | **81.83** | **78.45** | **73.07** |

Table 12: Performance comparison between quantized and non-quantized models based on the Qwen 2.5 family with comparable main body memory usages.

| Method | Memory | Size | Bits | Wiki2↓ | C4↓ | ArcC↑ | ArcE↑ | HellaS↑ | PiQA↑ | WinoG↑ | Acc↑ |
|--------|--------|------|------|--------|-----|-------|-------|---------|-------|--------|------|
| - | 2.62GiB | 1.5B | 16 | **8.58** | 12.54 | 41.21 | 75.51 | 50.12 | 75.79 | 63.85 | 61.30 |
| GPTQ | 2.45GiB | 7B | 3 | 10.73 | 13.90 | 39.33 | 71.21 | 50.16 | 73.29 | 60.77 | 58.95 |
| GPTAQ | 2.45GiB | 7B | 3 | 11.44 | 14.28 | 40.1 | 70.88 | 51.32 | 74.54 | 62.51 | 59.87 |
| NeUQI | 2.45GiB | 7B | 3 | 8.64 | **11.72** | **44.80** | **76.77** | **55.15** | **77.97** | **70.96** | **65.13** |
| - | 5.55GiB | 3B | 16 | 7.44 | 11.15 | 45.22 | 77.31 | 55.03 | 78.62 | 68.75 | 64.99 |
| GPTQ | 4.96GiB | 14B | 3 | 8.48 | 11.61 | 38.82 | 67.21 | 54.22 | 75.52 | 65.51 | 60.26 |
| GPTAQ | 4.96GiB | 14B | 3 | 8.82 | 11.7 | 45.22 | 76.43 | 55.47 | 76.55 | 68.67 | 64.47 |
| NeUQI | 4.96GiB | 14B | 3 | **6.82** | **9.89** | **50.17** | **80.43** | **59.39** | **79.54** | **76.09** | **69.12** |
| - | 13.05GiB | 7B | 16 | 6.39 | 10.02 | 48.29 | 80.56 | 60.00 | 78.67 | 72.69 | 68.04 |
| GPTQ | 11.72GiB | 32B | 3 | 7.21 | 10.36 | 44.03 | 74.96 | 58.75 | 77.97 | 66.22 | 64.38 |
| GPTAQ | 11.72GiB | 32B | 3 | 7.7 | 10.62 | 44.62 | 77.31 | 59.34 | 79.71 | 69.14 | 66.03 |
| NeUQI | 11.72GiB | 32B | 3 | **5.85** | **9.31** | **51.19** | **80.98** | **62.30** | **80.14** | **75.14** | **69.95** |
| - | 26.42GiB | 14B | 16 | **4.93** | 8.75 | 55.80 | 82.37 | 63.38 | 81.07 | 74.66 | 71.46 |
| GPTQ | 26.36GiB | 72B | 3 | 6.89 | 9.96 | 46.93 | 78.03 | 61.51 | 78.62 | 71.43 | 67.30 |
| GPTAQ | 26.36GiB | 72B | 3 | 7.94 | 9.93 | 50.09 | 80.68 | 61.8 | 79.71 | 70.4 | 68.54 |
| NeUQI | 26.36GiB | 72B | 3 | 4.99 | **8.37** | **56.40** | **83.75** | **64.92** | **81.83** | **78.45** | **73.07** |

