# OpenReview forum: "NeUQI: Near-Optimal Uniform Quantization Parameter Initialization"
_ICLR.cc/2026/Conference — Submitted to ICLR 2026_

### Official Review · Reviewer_krwb · 2025-10-31

**Soundness:** 3
**Presentation:** 3
**Contribution:** 2
**Rating:** 6
**Confidence:** 5

**Summary:**

This paper addresses the problem of parameter initialization in uniform post-training quantization (PTQ) of large language models (LLMs). While most prior works rely on the conventional Min-Max strategy, which becomes ineffective at low bit-widths (2–3 bits), the authors identify two key limitations of this rationale: (1) scale and zero-point are tied to extreme values, unnecessarily enlarging the search space, and (2) the integer constraint on zero-point reduces flexibility. To overcome these, they propose NeUQI, a near-optimal initialization method that directly optimizes scale and zero-point using a loss-aware formulation with efficient algorithms. They further combine NeUQI with a lightweight distillation strategy. Experiments on LLaMA and Qwen families show consistent improvements, with NeUQI sometimes even surpassing non-quantized baselines at comparable memory usage.

However, some limitations remain. First, while floating-point zero-points could in principle offer better representational flexibility than integer zero-points, many hardware platforms do not support them, limiting practical applicability. Second, the range of baseline initialization methods evaluated is still limited.

**Strengths:**

- Strong theoretical grounding with efficient algorithms (increment-prefix-sum, coarse-to-fine scale search).
- Extensive experiments across multiple LLM families. The experiments discuss the combination of this initialization method with various quantization algorithms, with consistent improvements.

**Weaknesses:**

In model quantization, it is natural to expect that using a floating-point zero-point would outperform an integer zero-point, since the representational capability is inherently greater. However, many hardware platforms do not support zero-points in floating-point formats, so such zero-point optimizations do not always hold in practice.

**Questions:**

The experiments of initialization methods considered is still quite limited. The author’s mentioned approaches, such as clipping-based, quantile-based, and Mean-Std-based methods, were not compared, nor were classical techniques like KL-divergence minimization. These do not conflict with the scale-learnable methods discussed by the author, such as OmniQuant and LearnQuant; combining these classical initialization methods with learnable quantization parameters , would perform worse than the author’s approach?


It is generally believed that when using fine-tuned or LoRA fine-tuned models, the advantage of initialization is largely diminished, since the weights adapt to the scale and zero-point. Does this perspective still apply to your proposed quantization initialization method?

---

> ### Author Response · Authors · 2025-11-21
> **Rebuttal to Reviewer krwb**
>
> Thank you for your valuable comments and suggestions. Our detailed responses to each concern are given below.
>
> > W: In model quantization, it is natural to expect that using a floating-point zero-point would outperform an integer zero-point, since the representational capability is inherently greater. However, many hardware platforms do not support zero-points in floating-point formats, so such zero-point optimizations do not always hold in practice.
>
> **A:** We fully agree that a floating-point zero-point is theoretically expected to outperform an integer zero-point due to its inherently greater representational capability. However, finding the near-optimal solution is challenging, and our NeUQI method is designed to address this challenge.
>
> For hardware support, the BitBLAS library [1], which has been integrated into the commonly used inference framework vLLM, supports GPUs such as A100 and RTX 4090, and also supports zero-points in floating-point, as documented in the BitBLAS Python API [2] when the `zeros_mode` is set to `original`. This demonstrates that our proposed approach has been supported by mainstream GPU kernels such as those in BitBLAS, without requiring specialized hardware modifications.
>
>
> > Q1: The experiments of initialization methods considered is still quite limited. The author’s mentioned approaches, such as clipping-based, quantile-based, and Mean-Std-based methods, were not compared, nor were classical techniques like KL-divergence minimization. These do not conflict with the scale-learnable methods discussed by the author, such as OmniQuant and LearnQuant; combining these classical initialization methods with learnable quantization parameters , would perform worse than the author’s approach?
>
> **A:** The initialization methods we selected, LeanQuant and Int-Search, already represent the strongest and most general search-based strategies available, and their search spaces inherently cover clipping-based, quantile-based, and Mean–Std-based methods. Therefore, combining them with these classical approaches is not applicable, as those methods are already evaluated within the optimization process. As for OmniQuant, it introduces Learnable Weight Clipping, which inherently integrates the clipping-based strategy into its optimization process and therefore does not require additional comparison. Regarding KL-divergence minimization, in low–bit-width settings such as 2-bit quantization, the quantized value space becomes too sparse to support stable KL-based optimization. For the same reason, KL-divergence minimization has primarily been applied in INT8 quantization. In the revised version, we further note that these classical initialization methods typically rely on empirical formulas, helping readers understand their underlying rationale.
>
>
> > Q2: It is generally believed that when using fine-tuned or LoRA fine-tuned models, the advantage of initialization is largely diminished, since the weights adapt to the scale and zero-point. Does this perspective still apply to your proposed quantization initialization method?
>
> **A:**  In low-bit regimes, the benefit of the better initialization is more evident compared with the high-bit setting, as it provides a stronger starting point that allows fine-tuning to more effectively preserve and enhance model performance. To further demonstrate the improvement of better initialization, we supplement the comparisons between the very strong fine-tuning method, EfficientQAT [3], and a modified version that replaces only the quantization parameter initialization with NeUQI. The results under 2-bit quantization with a group size of 128 on LLaMA 2 7B and LLaMA 3 8B are shown below. The results indicate that even for such a very strong fine-tuning method, NeUQI still brings further performance improvements. In the revised version, we have added the supplementary experiments with EfficientQAT to relevant sections, alongside the existing comparisons with PV-tuning, to further emphasize the importance of better initialization and the effectiveness of NeUQI.
>
>
> | Model       | Method               | PPL on Wiki2 |  PPL on C4   | Average accuracy on 5 benchmarks |
> |-------------|----------------------|-------|-------|------|
> | **LLaMA 2 7B** | EfficientQAT          | 7.19 | 8.79 | 59.50 |
> |              | NeUQI+EfficientQAT    | **6.81** | **8.41** | **60.01** |
> | **LLaMA 3 8B** | EfficientQAT          | 9.80 | 13.22 | 59.37 |
> |              | NeUQI+EfficientQAT    | **9.14** | **12.51** | **60.68** |
>
>
> [1] https://github.com/microsoft/BitBLAS
>
> [2] https://github.com/microsoft/BitBLAS/blob/main/docs/PythonAPI.md
>
> [3] Chen, Mengzhao, et al. "Efficientqat: Efficient quantization-aware training for large language models." Proceedings of the 63rd Annual Meeting of the Association for Computational Linguistics (Volume 1: Long Papers). 2025.

---

> ### Author Response · Authors · 2025-11-26
> **Warm Appreciation and Invitation for Further Feedback**
>
> Dear Reviewer krwb,
>
> We hope this message finds you well. Thank you for your valuable questions. We have carefully prepared our responses. We hope you will consider raising your score if our responses effectively addressed your questions.
>
> Best regards,
>
> Authors of NeUQI

---

### Official Review · Reviewer_cpcn · 2025-11-01

**Soundness:** 1
**Presentation:** 1
**Contribution:** 2
**Rating:** 2
**Confidence:** 3

**Summary:**

This paper addresses the 2-bit Post-Training Quantization problem for LLMs. The proposed method adopts a floating-point zero point for more precise optimization on the approximated quantization loss of layer-wise outputs. An efficient searching algorithm is proposed to find the optimal zero point and scaling.

**Strengths:**

- The proposed method shows robustness across different models and tasks in the 2-bit regime, surpassing SOTA performance.

**Weaknesses:**

- The definition of Min-Max initialization should be mentioned before/at Section 3, e.g., as a reference to Appendix A. I find it difficult to understand Section 3 at first without acknowledging the definition of what is referred to as "Min-Max" initialization.
- In Line 123, the grid search size $T$ is not properly defined/introduced. In particular, the definition in Appendix A does not mention about $T$.
- There is a typo in Eq. (3): $(W_{i,:}) - W_{i,:})^\top$ should be $(Q(W_{i,:}) - W_{i,:})^\top$
- The statement in Line 218 is not clearly written. I assume that stated example is referring to the scenario where $\exists z, z'$ such that ${\rm clip}(\lfloor x_i + z \rceil, 0, 2^k - 1) = a$ and ${\rm clip}(\lfloor x_i + z' \rceil, 0, 2^k - 1) = a+1$. Then, the increment in loss function should be written as $h_i((x_i + z' - (a+1))^2 - (x_i + z - a)^2 )$.

Overall, I think the optimization process in Section 4.3.1 and 4.3.2 should be described more clearly. Judging from the current submission, it is not clear how the proposed algorithm is implemented.

**Questions:**

- When the zero point optimization is simplified by the smoothed objective in Eq. (6), i.e., reducing the candidate solutions of $z$ to $\{1/2 - x_i, 2^k - 1/2 -x_i \}$, the rounded values are in $\{1/2, 2^k - 1/2 \}$. Is it equivalent to falling back to the **integer** zero point model $Q_{s, z}(x) = s\cdot ({\rm clip}(\lfloor x / s \rceil+ z , 0, 2^k - 1) - z)$, as opposed to the floating-point zero point model that is proposed in Section 4.1? I suggest the authors to clarify the advantage of adopting a floating-point $z$ when solving for the optimization in Section 4.3.1.

- In Line 10 of Algorithm 1, which coefficients are initialized and how are they initialized?

- How does Algorithm 1 ensures to find the optimal solution to Eq. (5), which involves the quantization error of a group of weight values $w_i$ (summation over $i = 1, ..., n$), using only a per-sample loss changes in Line 5 of Algorithm 1?

- Can the authors provide the actual time cost of performing NeUQI, and compare it to every baseline methods?

---

> ### Author Response · Authors · 2025-11-21
> **Rebuttal to Reviewer cpcn (Part 1/2)**
>
> Thank you for your valuable comments and suggestions. Our detailed responses to each concern are given below.
>
> > W1: The definition of Min-Max initialization should be mentioned before/at Section 3, e.g., as a reference to Appendix A. I find it difficult to understand Section 3 at first without acknowledging the definition of what is referred to as "Min-Max" initialization.
>
> **A:** Thank you for your helpful suggestions. In the revised version, we added a Background section and moved the original Section 3 and Section 4.1 into it, allowing readers to understand the Min–Max initialization before the later technical details.
>
> > W2: In Line 123, the grid search size $T$ is not properly defined/introduced. In particular, the definition in Appendix A does not mention about $T$.
>
> **A:** The grid search size $T$ refers to the grid search size used in LeanQuant, as mentioned in the preceding context, for searching the scale and zero-point. We have revised the corresponding part and explicitly referred to LeanQuant again to avoid any ambiguity.
>
> > W3: There is a typo in Eq. (3): $(W_{i,:})-W_{i,:})^\top$ should be $(Q(W_{i,:})-W_{i,:})^\top$
>
> **A:** Thank you for pointing this out. In the revised version, we have fixed the typo.
>
> > W4: The statement in Line 218 is not clearly written. I assume that stated example is referring to the scenario where $\exists z, z'$ such that $\text{clip}(\lfloor x_i + z \rfloor, 0, 2^k - 1) = a$ and $\text{clip}(\lfloor x_i + z' \rceil, 0, 2^k - 1) = a + 1$. Then, the increment in loss function should be written as $h_i((x_i + z' - (a + 1))^2 - (x_i + z - a)^2)$.
>
> **A:** As noted at the beginning of the paragraph, $\mathrm{clip}(\lfloor x_i + z \rceil, 0, 2^k - 1)$ is a piecewise constant function of $z$. Therefore, within this paragraph, $z$ should be treated as the variable. When $z$ increases and crosses a transition point, the function value changes from $a$ to $a + 1$. As shown in Line 5 of Algorithm 1, this transition occurs when $z$ reaches $t = j + \frac{1}{2} - x_i$, where the function value changes from $j$ to $j + 1$. In the revised version, we have added the phrase "when $z$ increases to cross the transition point" to explicitly indicate when this change occurs.
>
> > Overall, I think the optimization process in Section 4.3.1 and 4.3.2 should be described more clearly. Judging from the current submission, it is not clear how the proposed algorithm is implemented.
>
> **A:** In the revised version, we have updated Section 4.3 (now Section 4.2) to clarify the optimization procedures for scale and zero-point, making the proposed algorithm easier to follow.

---

> ### Author Response · Authors · 2025-11-21
> **Rebuttal to Reviewer cpcn (Part 2/2)**
>
> > Q1: When the zero point optimization is simplified by the smoothed objective in Eq. (6), i.e., reducing the candidate solutions of $z$ to $1/2 - x_i, 2^k - 1/2 - x_i$, the rounded values are in $1/2, 2^k - 1/2$. Is it equivalent to falling back to the **integer** zero point model $Q_{s,z}(x) = s \cdot (\text{clip}(\lfloor x/s \rfloor + z, 0, 2^k - 1) - z)$, as opposed to the floating-point zero point model that is proposed in Section 4.1? I suggest the authors to clarify the advantage of adopting a floating-point $z$ when solving for the optimization in Section 4.3.1.
>
> **A:** We would like to clarify two key misunderstandings in the comment:
> (1) $1/2 - x_i$ and $2^k - 1/2 - x_i$ are transition points of the function of $z$ in Eq. (6) (now Eq. (7)), shown in Algorithm 2 of Appendix C, rather than candidate solutions of $z$.
> (2) Section 4.3.1 Optimization of zero-point (now Section 4.2.1) does not include any statement about the rounded zero-points or values, and the only rounding operation involved is in computing the quantized value $\text{clip}(\lfloor x_i + s \rceil, 0, 2^k - 1)$.
>
> Furthermore, if the zero-point $z$ is restricted to $k$-bit unsigned integers as in previous work, the optimization can be performed by directly enumerating all possible integer values of $z$ to identify the optimal one, making it much simpler than when $z$ is treated as a floating-point variable.
>
> > Q2: In Line 10 of Algorithm 1, which coefficients are initialized and how are they initialized?
>
> **A:** Since $z$ is the variable of optimization, we initialize the quadratic form $A z^2 - 2B z + C$ using the coefficients obtained by expanding $\sum_i h_i (x_i + z)^2$.
>
> > Q3: How does Algorithm 1 ensures to find the optimal solution to Eq. (5), which involves the quantization error of a group of weight values $w_i$ (summation over $i = 1, \ldots, n$), using only a per-sample loss changes in Line 5 of Algorithm 1?
>
> **A:**
> Orginal Eq. (5) (now Eq. (6)):
> $$\mathcal{L}(z) = s^2\sum_{i=1}^n h_i \big( \text{clip}\left(\lfloor x_i + z \rceil, 0, 2^k - 1\right)  - (x_i + z) \big)^2$$
> Since Eq. (5) is the sum of piecewise quadratic per-sample losses, it is itself a piecewise quadratic function, and its transition points are the union of those of the per-sample losses, with identical coefficient changes. Therefore, Algorithm 1 starts from the coefficients in the first interval and derives those for all subsequent intervals based on the coefficient changes between adjacent intervals. With the coefficients of each interval thus obtained, the minimum within each interval can be computed in closed form, and the global minimum is simply the smallest among these interval-wise minima. In the revised version, we describe in detail the step that accumulates the coefficient changes to obtain the coefficients for each interval in the algorithm, helping readers better understand the procedure.
>
> > Q4: Can the authors provide the actual time cost of performing NeUQI, and compare it to every baseline methods?
>
> **A:**  The table below summarizes the quantization process runtime of all baseline methods and NeUQI applied to the LLaMA-2 family under 2-bit channel-wise quantization on a single NVIDIA A40 GPU. Through our algorithmic design, NeUQI maintains a well-balanced quantization runtime while achieving considerable performance improvements, which facilitates further research and exploration. Moreover, the quantization process directly produces **a fully quantized model ready for immediate use**, without incurring any additional inference-time overhead.
>
>
> | Method        | 7B (minutes) | 13B (minutes) | 70B (minutes) |
> | ------------- | -------- | --------- | --------- |
> | GPTQ          | 8.75     | 15.70     | 76.72     |
> | GPTAQ         | 14.35    | 27.23     | 178.08    |
> | MagR          | 21.87    | 49.70     | 396.45    |
> | NeUQI         | 29.75    | 53.03     | 292.93    |

---

> ### Author Response · Authors · 2025-11-26
> **Warm Appreciation and Invitation for Further Feedback**
>
> Dear Reviewer cpcn,
>
> We hope this message finds you well. Thank you for your valuable questions. We have carefully prepared our responses. We hope you will consider raising your score if our responses effectively addressed your questions.
>
> Best regards,
>
> Authors of NeUQI

---

> > ### Comment · Reviewer_cpcn · 2025-11-28
> >
> > I have read the revised submission. My main concern remains on the presentation of the proposed method, especially Section 4.2.1. For instance, it is still not clear to me how to interpret the loss increment $h_i((x_i + z - (a+1))^2 - (x_i + z - a)^2 )$, and why it is not $h_i((x_i + z' - (a+1))^2 - (x_i + z - a)^2 )$ for two different values of $z', z$ as I have mentioned in my first review. May the authors explain further, especially what does the value $a$ stands for in this equation?

---

> ### Author Response · Authors · 2025-12-03
>
> We apologize for not explaining the meaning of the formula clearly. Actually, the increment $h_i((x_i+z-(a+1))^2 - (x_i+z-a)^2)$ represents the difference between the quadratic functions of $z$ on two adjacent intervals around the transition point $a+\tfrac12 - x_i$ for the per-sample loss $\mathcal{L}_i(z)$, which is a piecewise quadratic function. In this expression, $z$ should be understood not as a fixed numerical value but as the shared independent variable.
>
> Based on your further valuable feedback, we realized that our previous description of the piecewise quadratic function of $z$ may lead to certain confusion about $z$ for readers.
>
> Therefore, we have carefully revised Section 4.2.1 Optimization of Zero-point. In the revised version, to explicitly show that $\mathcal{L}_i(z)$ is a piecewise quadratic function of $z$ and to specify the quadratic function on each interval, we rewrite $\mathcal{L}_i(z)$ in an explicit piecewise-defined form.
>
> To ensure that the introduction of the acceleration method is not abrupt, we added a guiding explanation. We first describe the naive method, which enumerates all intervals and sums the corresponding per-sample quadratic functions on each interval. The complexity of the naive method is prohibitively large, so we need an accelerated approach. We then note that between two adjacent intervals, only a single quadratic function in the sum changes. Consequently, we obtain the accelerated method: the sum on the current interval equals the sum on the previous interval plus the corresponding change in that quadratic function, for example $\delta_{i,j}(z)=h_i(x_i+z-(j+1))^2-h_i(x_i+z-j)^2$.

---

### Official Review · Reviewer_2BFd · 2025-11-01

**Soundness:** 3
**Presentation:** 3
**Contribution:** 3
**Rating:** 4
**Confidence:** 3

**Summary:**

This paper presents NeUQI, a parameter initialization algorithm for uniform quantization in post-training quantization (PTQ) of large language models. The authors identify two inherent limitations in the widely used Min–Max initialization strategy — its reliance on extreme values and integer zero-point constraints — and propose a new method that relaxes these assumptions. NeUQI directly optimizes scale and zero-point via a loss-aware formulation and an efficient two-stage approximation algorithm, showing consistent improvements across LLaMA and Qwen model families. The method also integrates effectively with lightweight distillation, surpassing heavier fine-tuning baselines such as PV-tuning.

**Strengths:**

The paper highlights the role of parameter initialization — and convincingly argues that initialization quality strongly affects quantization performance. By systematically reformulating the initialization problem and relaxing integer constraints, the proposed NeUQI provides an efficient and theoretically grounded approach. The experimental results are competitive. The inclusion of lightweight distillation further enhances the practical significance of the method.

**Weaknesses:**

However, the empirical evidence for the independent importance of initialization remains somewhat ambiguous. Many PTQ methods that incorporate limited fine-tuning (e.g., OmniQuant) already perform a form of implicit re-initialization by adjusting parameters during optimization. It would therefore be more convincing if the paper explicitly compared NeUQI with such fine-tuning-based methods across a wider range of models. Although Table 1 includes some results for OmniQuant and PV-tuning, the current coverage is limited. Additional experiments on more architectures, and under both weight-only and weight-activation quantization, would strengthen the claim that NeUQI surpasses these adaptive methods in a consistent and generalizable way.

**Questions:**

see above.

---

> ### Author Response · Authors · 2025-11-21
> **Rebuttal to Reviewer 2BFd**
>
> Thank you for your valuable comments and suggestions. Our detailed responses to each concern are given below.
>
> In the "Distillation" paragraph of Section 5.5 Discussion (now the "Distillation and Fine-tuning" paragraph), we clearly emphasize the importance of better initialization within fine-tuning methods, rather than claiming the independent importance of initialization. We suggest that interpreting fine-tuning as an "implicit re-initialization" oversimplifies its role. Ideally, initialization defines the starting point for quantization-aware optimization, whereas fine-tuning directs the subsequent optimization trajectory. They serve different purposes and follow different optimization strategies. As shown in Table 5, the advantage of better initialization is clear: NeUQI, when combined with lightweight distillation, already surpasses PV-tuning, even though the latter requires substantially greater computational resources and involves a much broader fine-tuning scope.
>
> To further demonstrate the advantage of the better initialization within fine-tuning methods, we supplement the comparisons between the very strong fine-tuning method, EfficientQAT [1], and a modified version that replaces only the quantization parameter initialization with NeUQI. The results under 2-bit quantization with a group size of 128 on LLaMA 2 7B and LLaMA 3 8B are shown below. The results indicate that even for such a very strong fine-tuning method, NeUQI still brings further performance improvements. In the revised version, we have added the supplementary experiments with EfficientQAT to relevant sections, alongside the existing comparisons with PV-tuning, to further emphasize the importance of better initialization and the effectiveness of NeUQI.
>
>
> | Model       | Method               | PPL on Wiki2 |  PPL on C4   | Average accuracy on 5 benchmarks |
> |-------------|----------------------|-------|-------|------|
> | **LLaMA 2 7B** | EfficientQAT          | 7.19 | 8.79 | 59.50 |
> |              | NeUQI+EfficientQAT    | **6.81** | **8.41** | **60.01** |
> | **LLaMA 3 8B** | EfficientQAT          | 9.80 | 13.22 | 59.37 |
> |              | NeUQI+EfficientQAT    | **9.14** | **12.51** | **60.68** |
>
> [1] Chen, Mengzhao, et al. "Efficientqat: Efficient quantization-aware training for large language models." Proceedings of the 63rd Annual Meeting of the Association for Computational Linguistics (Volume 1: Long Papers). 2025.

---

> ### Author Response · Authors · 2025-11-26
> **Warm Appreciation and Invitation for Further Feedback**
>
> Dear Reviewer 2BFd,
>
> We hope this message finds you well. Thank you for your valuable questions. We have carefully prepared our responses. We hope you will consider raising your score if our responses effectively addressed your questions.
>
> Best regards,
>
> Authors of NeUQI

---

### Official Review · Reviewer_gwSK · 2025-11-01

**Soundness:** 3
**Presentation:** 2
**Contribution:** 3
**Rating:** 6
**Confidence:** 4

**Summary:**

This paper addresses the deployment challenges of large language models (LLMs) on resource-limited devices caused by their high memory and computation demands. It focuses on post-training quantization (PTQ) with uniform quantization, which is hardware-friendly and widely supported. The authors observe that prior work has improved quantization techniques but largely neglected how quantization parameters are initialized, which typically relies on the suboptimal Min–Max strategy. To overcome this, they propose NeUQI, an efficient method for finding near-optimal initialization of uniform quantization parameters. Experiments on LLaMA and Qwen models show that NeUQI consistently outperforms existing PTQ approaches.

**Strengths:**

- The paper identifies and tackles an underexplored but critical aspect of post-training quantization: the initialization of quantization parameters (i.e., scaling-factor and zero-point). By moving beyond the traditional Min-Max initialization, the proposed NeUQI method provides a principled and efficient way to find near-optimal initialization, leading to consistent performance gains across multiple LLMs.
- Extensive experiments on LLaMA and Qwen model families demonstrate that NeUQI achieves SOTA results in various settings.

**Weaknesses:**

- Some methodological details are missing. In Eq. (3), the quantized weights appear not to be explicitly optimized, the loss function only involves the scaling factor and zero-point. In contrast, GPTQ explicitly updates weights through a reconstruction step. Please clarify how the quantized weights are incorporated into the optimization process and whether they are updated similarly.
- The paper lacks an ablation study evaluating the effect of using a floating-point zero-point. Since this design choice deviates from conventional integer quantization schemes, an analysis of its contribution to performance would strengthen the paper.
- The writing could be improved for clarity and accessibility. In the Abstract and Introduction, the description remains overly high-level. For instance, mentioning the replacement of the Min–Max scheme with a “better initialization” without concrete details. As a result, readers cannot grasp the essence of the proposed method until well into Section 4. It is recommended to summarize the key technical innovations and intuitions earlier to enhance readability and highlight the paper’s novelty.

**Questions:**

- Is the lightweight distillation strategy incorporated into NeUQI in Table 3?
- Can the proposed method be extended to MoE-based models?

---

> ### Author Response · Authors · 2025-11-21
> **Rebuttal to Reviewer gwSK**
>
> Thank you for your valuable comments and suggestions. Our detailed responses to each concern are given below.
>
> > W1: Some methodological details are missing. In Eq. (3), the quantized weights appear not to be explicitly optimized, the loss function only involves the scaling factor and zero-point. In contrast, GPTQ explicitly updates weights through a reconstruction step. Please clarify how the quantized weights are incorporated into the optimization process and whether they are updated similarly.
>
> **A:** It is important to note that our NeUQI is a method dedicated to initializing the quantization parameters $s$ and $z$, rather than optimizing the weights. Therefore, in our formulation, $s$ and $z$ are naturally treated as the sole optimization targets. The quantized weights are determined in the following quantization step based on the initialized quantization parameters. In the revised version, we have added additional clarifications to avoid potential misunderstandings.
>
> > W2: The paper lacks an ablation study evaluating the effect of using a floating-point zero-point. Since this design choice deviates from conventional integer quantization schemes, an analysis of its contribution to performance would strengthen the paper.
>
> **A:** We apologize for the misunderstanding, and actually the ablation results were included in the "Integer Constraint" paragraph of Section 5.4 Bit Analysis and Table 3. Except for NeUQI, all other methods are implemented with the integer constraint on the zero-points, and among them, Int-Search uses the same loss function as NeUQI. Furthermore, Table 4 provides additional confirmation of the effectiveness of this design choice. In the revised version, we have added additional clarifications to help readers locate the ablation study.
>
> > W3: The writing could be improved for clarity and accessibility. In the Abstract and Introduction, the description remains overly high-level. For instance, mentioning the replacement of the Min–Max scheme with a "better initialization" without concrete details. As a result, readers cannot grasp the essence of the proposed method until well into Section 4. It is recommended to summarize the key technical innovations and intuitions earlier to enhance readability and highlight the paper’s novelty.
>
> **A:** Thank you for your helpful suggestion. In the revised version, we have added the description of the algorithmic design of NeUQI in the Abstract and Introduction to help readers grasp its core idea more easily.
>
> > Q1: Is the lightweight distillation strategy incorporated into NeUQI in Table 3?
>
> **A:** No. In Table 3, we compare NeUQI with other initialization methods that still constrain zero-points to be integers, and lightweight distillation is not incorporated into NeUQI. The results of NeUQI with lightweight distillation are reported in Table 5 for comparison with other fine-tuned methods.
>
> > Q2: Can the proposed method be extended to MoE-based models?
>
> **A:** Yes. Similar to LeanQuant, GPTQ, and GPTAQ, our NeUQI operates at the level of individual linear layers and can be directly applied to the linear layers within MoE architectures in the same manner as existing quantization methods.

---

> ### Author Response · Authors · 2025-11-26
> **Warm Appreciation and Invitation for Further Feedback**
>
> Dear Reviewer gwSK,
>
> We hope this message finds you well. Thank you for your valuable questions. We have carefully prepared our responses. We hope you will consider raising your score if our responses effectively addressed your questions.
>
> Best regards,
>
> Authors of NeUQI

---

### Author Response · Authors · 2025-11-21
**Response to all Reviewers**

We thank all reviewers for their valuable efforts, and we have carefully addressed all concerns and revised the paper accordingly. Following the ICLR author guidelines, we used the additional page allowed during the discussion/rebuttal phase to incorporate the updates. All revised and supplementary content has been highlighted in **blue**.

---

### Author Response · Authors · 2025-12-03
**Summary of Contribution and Rebuttal**

Dear Area Chair,

We sincerely appreciate your time and dedication to the conference, especially under the current special and urgent circumstances, and we fully respect your evaluation of our study.

We would like to briefly outline the core contributions of our work.

+ We explore the initialization of quantization parameters, identify the limitations of traditional Min–Max methods, and propose NeUQI to overcome these limitations. (as noted by Reviewer gwSK and Reviewer 2BFd)
+ In moving beyond traditional Min–Max and addressing the optimization of scale and zero-point in the quantization loss, we propose an efficient and theoretically grounded solution. (as noted by Reviewer 2BFd and Reviewer krwb)
+ We conduct extensive experiments across multiple LLM families, and our method consistently improves performance, thereby thoroughly demonstrating its robustness and effectiveness. (as noted by Reviewer gwSK and Reviewer krwb)
+ Our method delivers strong performance across diverse models, tasks, and bit settings, particularly in the challenging 2-bit regime. (as noted by Reviewer gwSK, Reviewer 2BFd, and Reviewer cpcn)

We would also like to briefly outline the revisions made during the rebuttal phase to address the reviewers’ concerns and improve the overall quality of the paper.

+ We fixed typos and made minor refinements to the relevant paragraphs in Section 2 Quantization Parameter Initialization, Section 3.3, Section 4.1, and Section 5.4 Integer Constraint to avoid potential misunderstandings.
+ We added a Background section and integrated into it the material previously presented in Section 4.1 Uniform Quantization, the appendix discussion on Min–Max initialization, and the original Section 3 Limitation within Min–Max, in order to provide a clearer explanation of traditional Min–Max initialization at the beginning.
+ We updated the Abstract and Introduction with a clearer description of the algorithmic design of NeUQI, allowing readers to grasp its core idea more easily.
+ We expanded Section 4.2 to include a more detailed overview of the algorithm, carefully revised Section 4.2.1 to provide additional conceptual guidance and to avoid potential misunderstandings regarding the role of $z$, and made minor revisions to Section 4.2.2 to improve clarity.
+ We added a comparison between EfficientQAT with and without NeUQI. The results show that NeUQI further improves the performance of the strong fine-tuning method EfficientQAT.
+ We added an additional Appendix E, which discusses hardware support for floating-point zero-points and notes that our method is already supported by existing BitBLAS library kernels.


Best regards,

The Authors

---

### Meta-Review · Area_Chair_RgQK · 2026-01-08

**Summary:**

This paper focuses on post-training quantization (PTQ) with uniform quantization. All reviewers acknowledge that it makes sense to replace the conventional min-max initialization with learning the initialization parameters directly. The major review concerns are mainly about methodology details, writing issues, more comprehensive ablations, and comparisons. The rebuttal addressed most of the issues.

However, one major issue arose from the rebuttal. As indicated in the rebuttal to Reviewer cpcn Q4, the runtime of NeUQI is much longer than the fine-tuning methods. This raises the question whether the comparison results (e.g., Table 1) are fair or not, if the computations are different for different methods. In other words, the fine-tuning methods should be given the same amount of computing/fine-tuning time for optimization for fair comparisons. In addition, one reviewer also expressed significant concerns about the writing and presentations.

**Reviewer Concerns:**

See above.

**Reviewer Scores:**

Reviewer 2BFd might upgrade the score since his comments are addressed in the rebuttal, and he is not confident.

---

### Decision · Program_Chairs · 2026-01-26

Reject